# The link between Somalian Plate rotation and the East African Rift System: an analogue modelling study

Frank Zwaan[1,2,3], Guido Schreurs[1]

[1] University of Bern, Institute of Geological Sciences, Baltzerstrasse 1+3, 3012 Bern, Switzerland
5 [2] Helmholtz Centre Potsdam - GFZ German Research Centre for Geosciences, Telegrafenberg, 14473 Potsdam, Germany
[3] University of Fribourg, Department of Geosciences, Chemin du Musée 6, 1700 Fribourg, Switzerland

*Correspondence to*: Frank Zwaan (frank.zwaan@gfz-potsdam.de)

Keywords: East African Rift, Nubian Plate, Somalian Plate, Victoria Plate, rifting, rotational extension, continental breakup,
10 analogue modelling, monitoring techniques

**Short Summary:** The East African Rift System (EARS) is a major plate tectonic feature splitting the African continent apart. Understanding the tectonic processes involved is of great importance for societal and economic reasons (natural hazards, resources). Laboratory experiments allow us to simulate these large-scale processes, highlighting the links between rotational 15 plate motion and the overall development of the EARS. These insights are relevant when studying other rift systems around the globe as well.

**Abstract:** The East African Rift System (EARS) represents a major tectonic feature splitting the African continent apart into the Nubian Plate situated to the west, and the Somalian Plate to the east. The EARS comprises various rift segments and 20 microplates, and represents a key location for studying rift evolution. Researchers have proposed various scenarios for the evolution of the EARS, but the impact of continent-scale rotational rifting, linked to the rotation of the Somalian Plate, has received only limited attention. In this study we apply analogue models to explore the dynamic evolution of the EARS within the broader rotational rifting framework. Our models show that rotational rifting leads to the lateral propagation of deformation towards the rotation axis, which reflects the general southward propagation of the EARS, but we must distinguish between the 25 propagation of distributed deformation, which can move very rapidly, and localized deformation, which can significantly lag behind. The various structural weakness arrangements in our models (simulating the pre-existing lithospheric heterogeneities that localize rifting along the EARS) lead to a variety of structures. Laterally overlapping weaknesses are required for localizing parallel rift basins to create rift pass structures, leading to the rotation and segregation of microplates such as the Victoria Plate in the EARS. Additional model observations concern the development of early pairs of rift-bounding faults flanking the rift 30 basins, followed by the localization of deformation along the axes of the most developed rift basins. Furthermore, the orientation of rift segments with respect to the regional (rotational) plate divergence affects deformation along these segments: oblique rift segments are less wide due to a strike-slip deformation component. Overall, our model results generally fit the

large-scale present-day features of the EARS, with implications for general rift development, and for the segregation and rotation of the Victoria plate.

## 1. Introduction

The East African Rift System (EARS) represents a major tectonic feature that splits the African continent in two main plates: the Nubian Plate situated to the west, and the Somalian Plate to the east (e.g., Morley et al., 1999; Chorowicz, 2005; Ring, 2014; Saria et al. 2014; Macgregor, 2015; Fig. 1a, b). From its northern end in the Afar triangle, where the rift merges with the Red Sea and Gulf of Aden basins, the EARS stretches some 5000 km southward and comprising various active rift segments and branches, as well as a number of microplates (e.g., Ebinger, 1989; Chorowicz, 2005; Corti, 2012; Stamps et al. 2012, 2021; Saria et al., 2013, 2014; Daly et al., 2020; Michon et al., 2022, Fig. 1a, b). The EARS is a key location for the study of rift evolution as it contains rift basins in various stages of development, from young continental rift basins to incipient continental break-up situations (e.g., Chorowicz, 2005; Ebinger, 2005; Corti, 2012, Macgregor 2015; Fig. 1a, b).

The development of the present-day EARS is thought to have started some 28 Myr ago in the Turkana part of the Kenya Rift (Fig. 1b), following earlier phases of mantle-induced volcanism, and researchers have proposed various scenarios for the subsequent development of the various basins in the rift system (e.g., Chorowicz et al. 2005; Morley, 2010; Macgregor 2015; Purcell 2018; Glerum et al., 2020; Martin 2023). Some authors considered large-scale NW-SE plate divergence to have caused rifting along the various rift segments, with offsets between the segments accommodated by large-scale NW-SE oriented strike-slip zones (e.g., Wheeler and Karson, 1994; Scott et al., 1989; Chorowicz, 2005). Other authors have favoured general E-W plate divergence along the EARS (e.g., Lezzar et al. 2002; Morley 2010; Delvaux et al. 2012), which is in line with GPS observations (e.g. Calais et al., 2006; Saria et al. 2013, 2014; Stamps et al., 2021). In general, the localization of the rift basins is thought to be controlled by the arrangement of inherited lithospheric weaknesses such as the Proterozoic Pan-African structural grain or the Mesozoic Davie Fracture Zone, which are more readily reactivated during tectonic deformation than undisturbed or cratonic lithosphere (e.g., Ring, 2014; Kendall and Lithgow-Bertelloni, 2016; Phethean et al. 2016; Michon et al. 2020, and references therein). However, the exact timing of rift initiation in the different parts of the EARS remains debated, although the general trend is reported to be one of southward younging over the past 30 Myr (e.g., Chorowicz 2005; Macgregor 2015; Michon, 2020; Martin 2023, and references therein).

An important aspect of the present-day EARS is rotational plate motion, with Martin (2023) highlighting the opposite rotations of the Victoria and Rovuma plates (Fig. 1a). Numerical modelling efforts by Glerum et al. (2020) show that the counter-clockwise rotation of the Victoria plate occurs due to general E-W extension acting on the overlapping branches of the EARS (i.e. the Western Rift and Kenya Rift) that cause the Victoria Plate in between to rotate in a counter-clockwise fashion (a so-called "rift pass" structure, e.g. Oldenburg and Brune, 1975; Nelson et al., 1992; Hieronymus, 2004; Katz et al., 2005; Tentler

and Accocella, 2010; Brune et al., 2017; Zwaan et al., 2018a, Fig. 1c). Glerum et al. (2020) also demonstrate that mantle flow linked to mantle plume activity below the generally magma-rich EARS (e.g., Macgregor 2015; Michon et al. 2020, Rajaonarison et al., 2023, and references therein) is not required to drive Victoria Plate rotation, but that far-field tectonic deformation (i.e., passive rifting) suffices. Yet, the forces driving rifting along the EARS remain hotly debated (e.g., Kendall & Lithgow-Bertelloni, 2016; Rajaonarison et al. 2021; Martin, 2023, and references therein)

On a larger scale, we observe the present-day clockwise rotation of the Somalian Plate (e.g., Saria et al. 2013, 2014; Stamps et al., 2021), which has been occurring since at least 20 Ma (DeMets & Merkouriev, 2016, 2021), and the associated southward decrease in general E-W plate divergence along the EARS (Fig. 1a). Various tectonic modelling studies have shown that such plate divergence gradients have important effects on the evolution and propagation of rift systems (e.g., Souriot and Brun,

1992; Benes and Scott, 1996; Sun et al., 2009; Molnar et al., 2017, 2018; Farangitakis et al., 2019, 2021; Khalil et al., 2020; Zwaan et al., 2020; Schmid et al., 2022a). Although Glerum et al. (2020) did incorporate a gradual southward in-plate divergence velocity decrease into their models, they only focussed on the immediate surroundings of the Victoria plate (Fig. 1c). Recently, Zwaan and Schreurs (2020) presented a first-order analogue modelling study exploring the rift interaction structures in rotational and orthogonal rifting settings (Fig. 1d). Their model results suggested that large-scale rotational rifting

explained the large-scale southward younging trend of the rift system as reported by e.g., Chorowicz (2005) and Macgregor (2015). However, Zwaan and Schreurs (2020) focussed on systematically testing generic rift arrangements, and presented a qualitative model analysis based on visual inspection of time-lapse imagery, so that a comparison of their models to the EARS remains rather coarse (Fig. 1d). To our knowledge, no other large-scale (modelling) studies involving rotational rifting and focussing on the EARS exist.

Hence, we here present a new series of analogue tectonic models of rotational rifting that are specifically tailored to the tectonic setting in the EARS, with the aim to explore the dynamic evolution of the rift system in more detail. To this end we adopt the general plate configuration of the EARS and the rotational plate motion pattern of the Somalian Plate provided by GPS studies (e.g., Saria et al., 2013, 2014; Stamps et al., 2021), and plate reconstructions (DeMets & Merkouriev, 2016, 2021). We find

that our new models generally fit the large-scale features and present-day deformation of the EARS, with implications for general rift development, rift basin propagation, and segregation and rotation of the Victoria plate.

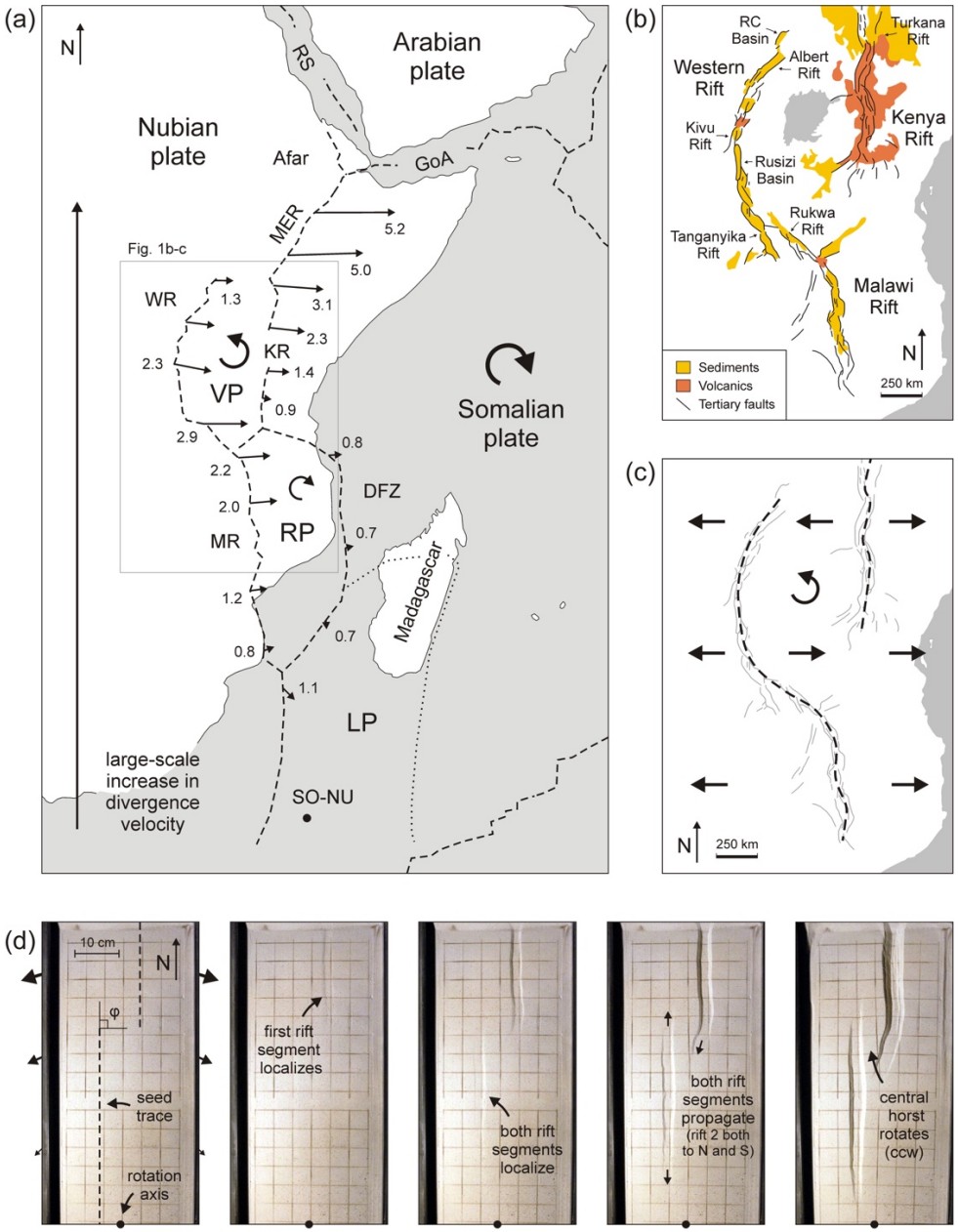

Figure. 1: (a) Tectonic outline of the East African Rift System (EARS) showing the rotation of the Somalian Plate, the Victoria plate (VP) and the Rovuma Plate (RP). Plate motions (in mm/yr), modified after Saria et al. (2013, 2014) and Stamps et al. (2021). DFZ: Davie Fracture Zone, GoA: Gulf of Aden, KR: Kenya Rift, LP: Lwandle Plate, MER: Main Ethiopian Rift, MR: Malawi Rift, RP: Rovuma Plate, RC: Rhino-Camp, RS: Red Sea, SO-NU: Somalia Nubia rotation pole, WR: Western Rift. (b) detailed structural map of the EARS. Modified after Ebinger (1989). (c). Tectonic models proposed for the EARS, after Glerum et al. (2020). (d) Analogue models of rift interaction in a rotational rifting setting by Zwaan and Schreurs (2020). Ccw: counterclockwise, φ: angle between the seed (i.e. inherited weakness) orientation and the line between both seed tips (90° in this case, i.e. no lateral seed overlap),

## 2. Methods

### 2.1. Set-up

In this study we apply a rotational extension set-up with a brittle-viscous layering, based on previous modelling efforts by
Zwaan and Schreurs (2020), Zwaan et al. (2020), and Schmid et al. (2022a, b) (Fig. 1). Prior to model preparation, an 8 cm
thick foam block (RG 50 polyurethane foam base from GOBAG, www.gobag.ch) is compressed between two longitudinal
sidewalls, on which the brittle-viscous model materials (1.5 cm quartz sand on top of a 3 cm thick viscous mixture representing
a 150 km thick lithosphere and the upper 300 km of the sub-lithospheric mantle, respectively) are applied. One of the sidewalls
is mobile and connected to a rotation axis, allowing it to rotate in a clockwise fashion, thus simulating the rotation of the
Somalian Plate with respect to the Nubian Plate, in a far-field tectonic setting (Fig. 1). Note that we do not distinguish between
continental and oceanic lithosphere in our models, assuming that the various plates along the EARS act as rigid blocks (Stamps
et al. 2021). During this rotational motion of the mobile sidewall the foam base expands, inducing extension in the overlying
model materials with a southwardly decreasing gradient (Fig. 2c, d). The rotational motion of the sidewall is set to a constant
divergence of 4 mm/h at the farthest point from the rotation axis, reproducing the relatively constant divergence of the Somalian
Plate about the present-day rotation pole since at least 20 Ma in the Horn of Africa (Saria et al. 2013, 2014; DeMets &
Merkouriev, 2016, 2021; Stamps et al. (2021). Given the model duration of 150 min, the total extension is 10 mm at the farthest
point from the rotation axis.

In order to localize deformation in our models, we include seeds that follow (parts of) the tectonic plate boundaries that form
the main present-day EARS template according to GPS analysis by Saria et al. (2013, 2014) and Stamps et al. (2021) (Figs.
1a, 2a-c, e). These seeds are thin (ø = 4 mm) semi-cylindrical bars of viscous material applied on top of the basal viscous
layer, causing the overlying sand layer to be locally thinned and weakened (Fig. 2b). Such seeds have been used in previous
modelling studies (e.g., Le Calvez & Vendeville, 2002; Zwaan et al., 2016; Molnar et al. 2019), and tend to produce strongly
localized rift basins and are deemed ideal to reproduce the localized style of deformation along pre-existing lithospheric
weaknesses in the EARS (e.g., Ring, 2014; Kendall and Lithgow-Bertelloni, 2016). We test five main combinations of seed
geometries to explore the effect different lithospheric weakness arrangements could have had on the evolution of the EARS,
and to obtain a best-fit model for comparison with the present-day rift system. Note that the seed representing the Western Rift
in Model D (Fig. 1, 2e) has a double thickness of 8 mm to represent a locally increased weakness in the lithosphere along that
part of the EARS, and that we do not include the Lwandle Plate (LP in Fig 1a) due to practical limitations. The summarized
results of one further experiment with an alternative rotation pole are presented in the Appendix. Detailed results regarding all
models are provided in the supplementary material (Zwaan and Schreurs, 2023).

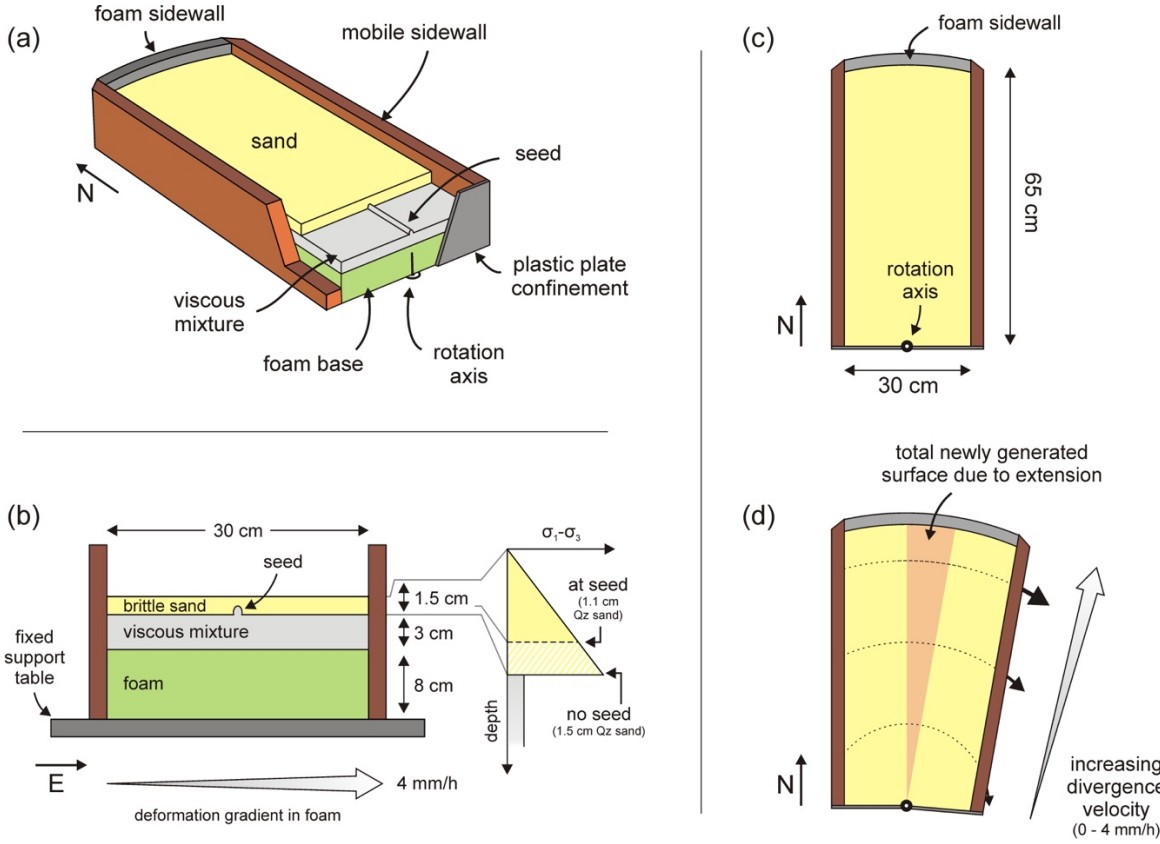

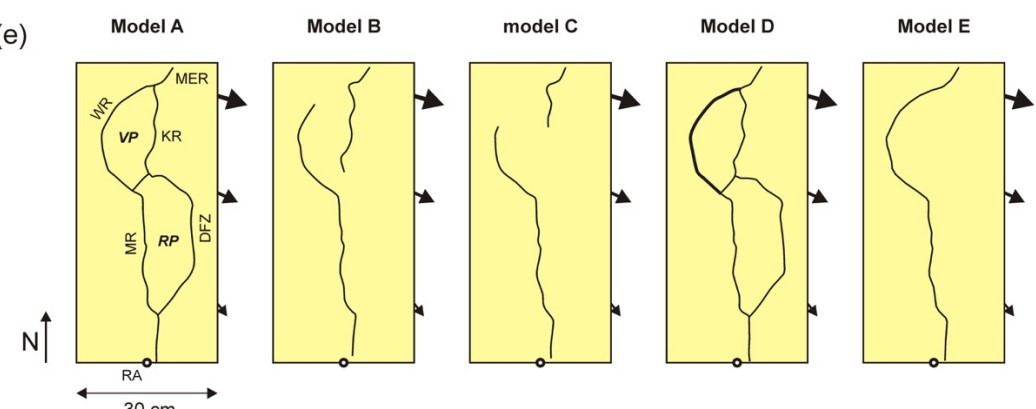

Figure 2: Model setup. (a) 3D sketch of model set-up. (b) Cross-section of model set-up. (c) top view of model set-up prior to deformation, and (d) after rotational deformation. (e) Seed geometries applied for Models A-E, where Model D contains a double thickness seed along the simulated Western Rift (WR). The seed geometries are based on the rift arrangement in the East African Rift System (EARS), as presented in Fig. 1a and indicated in the panel of Model A. DFZ: Davie Fracture Zone, KR: Kenya Rift, MER: Main Ethiopian Rift, MR: Malawi Rift, RP: Rovuma Plate, VP: Victoria Plate, WR: Western Rift. Note that the rotation axis (RA) is situated somewhat too much to the west when compared to the natural case (Fig. 1a), but this does not affect the model results in any significant fashion (see description of Model F in the Appendix).

## 2.2. Materials

We use a brittle-viscous model layering with a 1.5 cm thick top layer consisting of quartz sand representing a 150 km lithosphere dominated by brittle deformation. This quartz sand (Quarzsand A from Carlo Bernasconi AG, www.carloag.ch)
has a grain size of 60-250 µm and internal friction angles between 31.4˚-36.1˚, with a cohesion value of 9 Pa (Zwaan et al., 2018b) (Table 1). The sand has a constant density of 1560 kg/m$^3$ when sieved from ca. 30 cm height (Schmid et al., 2020), and is flattened by a scraper at the end of deformation to ensure a flat model surface.

The underlying 3 cm thick viscous layer representing the sub-lithospheric mantle consists of a near-Newtonian mixture of
150 SGM-36 Polydimethyl-siloxane (PDMS) and corundum sand (F120 Edelkorund) from Carlo AG ($\eta$ = ca. 1.5·105 Pa·s; n = 1.05-1.10, Zwaan et al., 2018c). The materials are mixed following a 0.965 : 1.00 weight ratio and the density of the resulting viscous mixture (ca. 1600 kg/m$^3$) is slightly higher than that of the sand layer. Further material properties are presented in Table 1.

**Table 1: Model materials**

| Granular materials | Quartz sand[a] | Corundum sand[b] |
|---|---|---|
| Grain size range | 60-250 µm | 88-125 µm |
| Density (bulk material)[c] | 2650 kg/m$^3$ | 3950 kg/m$^3$ |
| Density (sieved) | 1560 kg/m$^3$ | 1890 kg/m$^3$ |
| Angle of internal peak friction | 36.1˚ | 37˚ |
| Angle of dynamic-stable friction | 31.4˚ | 32˚ |
| Angle of reactivation friction | 33.5˚ | - |
| Cohesion | 9 ± 98 Pa | 39 ± 10 Pa |
| **Viscous materials** | **Pure PDMS[a, d]** | **PDMS / corundum sand mixture[a]** |
| Weight ratio PDMS / corundum sand | - | 0.965 kg / 1.00 kg |
| Density | 965 kg/m$^3$ | 1600 kg/m$^3$ |
| Viscosity | ca. 2.8·10$^4$ Pa·s | ca. 1.5·10$^5$ Pa·s[e] |
| Rheology[f] | Newtonian (n = 1) | near-Newtonian (n = 1.05-1.10) |

[a] Quartz sand, PDMS and viscous mixture characteristics after Zwaan et al. (2016, 2018b, c; Schmid et al. 2020)
[b] Corundum sand characteristics after Panien et al. (2006)
[c] Specific densities after Carlo AG (2022)
[d] Pure PDMS rheology after Rudolf et al. (2016)
[e] Viscosity value holds for model strain rates < 10$^{-4}$ s$^{-1}$
[f] Power-law exponent $n$ (dimensionless) represents sensitivity to strain rate

## 2.3. Scaling

We use standard scaling methods to make sure our analogue models adequately represent the natural situation in the EARS (see scaling values in Table 2). Since brittle materials have strain rate-independent rheologies, the angle of internal friction of our quartz sand (36.1˚) is the main factor for scaling purposes. This value is very similar to values obtained from experimentally deformed rocks (31-38˚, Byerlee, 1978, Table 2) and is generally considered to be similar for brittle parts of the mantle as well (e.g. Brun 1999, 2002). We assume a simplified brittle-dominated lithosphere in our models, which is therefore appropriately represented by our quartz sand.

Scaling viscous materials becomes more intricate than scaling brittle materials due to the strain rate-dependent rheology of the former. Using the stress ratio between model and nature ($\sigma*$, convention: $\sigma* = \sigma_{model}/ \sigma_{nature}$): $\sigma*= \rho*\cdot h*\cdot g*$, where $\rho*$, $h*$ and $g*$ are density, length and gravity ratios, respectively (Hubbert, 1937; Ramberg, 1981) and the viscosity ratio ($\eta*$), we obtain the strain rate ratio $\dot{\varepsilon}*$ (Weijermars and Schmeling 1986): $\dot{\varepsilon}* = \sigma*/\eta*$. With the strain rate ratio we can subsequently compute the velocity and time ratios ($v*$ and $t*$): $\dot{\varepsilon}* = v*/h* =1/t*$. Assuming a low viscosity of $1\cdot10^{20}$ Pa·s for the sub-lithospheric mantle (e.g., Steinberger & Calderwood, 2006), our 4 mm/h divergence velocity scales up to ca. $1\cdot10^4$ mm/yr. This velocity is much higher than typical rift divergence rates along the EARS (e.g., Saria et al., 2014; Stamps et al., 2021, Fig. 1), and previous modelling efforts have shown that high extension rates can lead to increased brittle-viscous coupling and distributed extension or "wide rifting" (e.g. Brun, 1999; Zwaan et al., 2016, 2019). However, as long as the seeds localize deformation properly, and no indications of distributed rifting are observed (as is clearly the case in our models), the high divergence velocity in our models can be deemed acceptable for the purpose of our study.

In addition, we consider the dynamic similarity of the model to the natural example through the Rs ratio and the Ramberg number. We derive the dynamic similarity between the brittle sand layer and the natural lithosphere using the Rs ratio between the gravitational stress and the cohesive strength or cohesion C (Ramberg, 1981; Mulugeta, 1988): $R_s$ = gravitational stress/cohesive strength = $(\rho\cdot g\cdot h)/C$. When assuming a combined cohesion of 100 MPa for the lithosphere, together with the 9 Pa cohesion of our quartz sand, we obtain a $R_s$ of 26 for our models and 27 for the natural example. The Ramberg number $R_m$ considers the dynamic similarity scaling of viscous materials (Weijermars and Schmeling 1986): $R_m$ = gravitational stress/viscous strength = $(\rho\cdot g\cdot h^2)/(\eta\cdot v)$, and has a value of 85 for both the viscous mixture and the sub-lithospheric mantle in nature, respectively. Since both the $R_s$ and $R_m$ values of our models are quite similar to those in their natural equivalent, we consider our models reasonably well scaled for simulating large-scale continental rifting processes.

**Table 2: Scaling parameters**

|  |  | Model | Nature |
|---|---|---|---|
| **General parameters** | Gravitational acceleration (g) | 9.81 m/s$^2$ | 9.81 m/s$^2$ |
|  | Divergence velocity (v) | 1.1·10$^{-6}$ m/s* | 3.4·10$^{-7}$ m/s |
| **Brittle layer** | Material | Quartz sand | Lithosphere |
|  | Peak internal friction angle ($\varphi$) | 36.1˚ | 30-38˚ |
|  | Thickness (h) | 1.5·10$^{-2}$ m | 1.5·10$^{5}$ m |
|  | Density ($\rho$) | 1560 kg/m$^3$ | 2800 kg/m$^3$ |
|  | Cohesion I | 9 Pa | 1.5·10$^{8}$ Pa |
| **Viscous / ductile layer** | Material | PDMS / corundum sand mixture | Sub-lithospheric mantle |
|  | Thickness (h) | 3·10$^{-2}$ m | 3·10$^{5}$ m |
|  | Density ($\rho$) | 1600 kg/m$^3$ | 3300 kg/m$^3$ |
|  | Viscosity ($\eta$) | 1.5·10$^{5}$ Pa·s | 1·10$^{20}$ Pa·s |
| **Dynamic scaling values** | Brittle stress ratio ($R_s$) | 26 | 27 |
|  | Ramberg number ($R_m$) | 85 | 85 |

* maximum divergence velocity, away from the rotation axis

## 2.4. Model monitoring and analysis

We monitor the surface evolution of our models through time-lapse photography, by means of a camera set-up consisting of 3 aligned high-resolution Nikon D810 (36.3 MP) cameras (Zwaan et al. 2021, 2022; Schmid et al. 2022a, b). Top view images from the central camera together with a 4 x 4 cm surface grid made of corundum sand (<1 mm thick) provides a first-order insight into model evolution. Additional model surface analysis involves photogrammetry on time-lapse imagery from the other two obliquely oriented cameras with Agisoft Photoscan software (www.agisoft.com) to generate digital elevation models, enabling an assessment of model topography variations over time, notably rift basin evolution, using QGIS software (www.qgis.org), (Fig. 4b). The time-lapse imagery also allows a more detailed analysis and quantification of model surface deformation through means of digital image correlation (DIC) techniques (e.g. Adam et al., 2005, Boutelier et al. 2019, and references therein). This DIC analysis involves the tracing of horizontal displacements between different time steps (here 10 mm of divergence) using LaVision DaVis 10.2 DIC software. From the horizontal displacement data we derive incremental and cumulative maximum normal strain maps that serve as a proxy for tracing active extensional deformation in the models over time.

## 3. Results

### 3.1. Model A

The results of Model A, with standard thickness seeds (ø = 4 mm), tracing most present-day, active EARS plate boundaries (see section 2.1), are presented in Fig. 3. Topography data show that early deformation is not clearly expressed when compared to what is revealed in the DIC results (as is the case in all subsequent models as well, Figs. 4-7). The first hints of topographic deformation appear in the northernmost part of the model after some 20 to 30 minutes, at the simulated Main Ethiopian Rift – Kenya rift location (Fig 3b, c). Subsequently a mostly N-S oriented through-going rift system develops as rift-related subsidence propagates southward, crossing over into the simulated Malawi Rift. Minor subsidence is visible at the simulated Western Rift as well, segregating the Victoria Plate model equivalent, but there is no trace of deformation along the eastern margin of the simulated Rovuma Plate (Fig. 3).

In contrast to the topography data, the DIC results show that deformation was in fact already localizing during the earliest stages of the model run (Fig. 3g, m). After 10 minutes, extensional deformation was developing along large parts of the most central seed segments. These structures develop conjugate sets of rift boundary faults as indicated by the parallel bands of localized deformation, and all structures active in Model A are seen to be deforming by the 20-minute mark (Fig. 3h, n) when the simulated Western Rift is also being active to a moderate degree, the effect of which is also nicely visible in the eastward displacement results (Fig. 3n, o). Furthermore, the clear northward increase in eastward displacement seen in the eastern part of the model highlights the clockwise rotational motion applied to the system (Fig. 3m-r). Note also the much less expressed gradient on the western side of the model, indicating some minor rotation there as well, which we regard to be a boundary effect (Fig. 3m-r).

Some additional important details are revealed by the DIC results. Firstly, we find that the southward propagation of active deformation is indeed largely completed some 20 minutes, but it continues at a slow rate until the end of the model run (Fig. 3g-l). Furthermore, towards the end of the model run, the most advanced rift basins in the model (the simulated Kenya and Malawi rifts) seem to develop a central band of extensional deformation in addition to the active rift boundary faults (Fig. 3l).

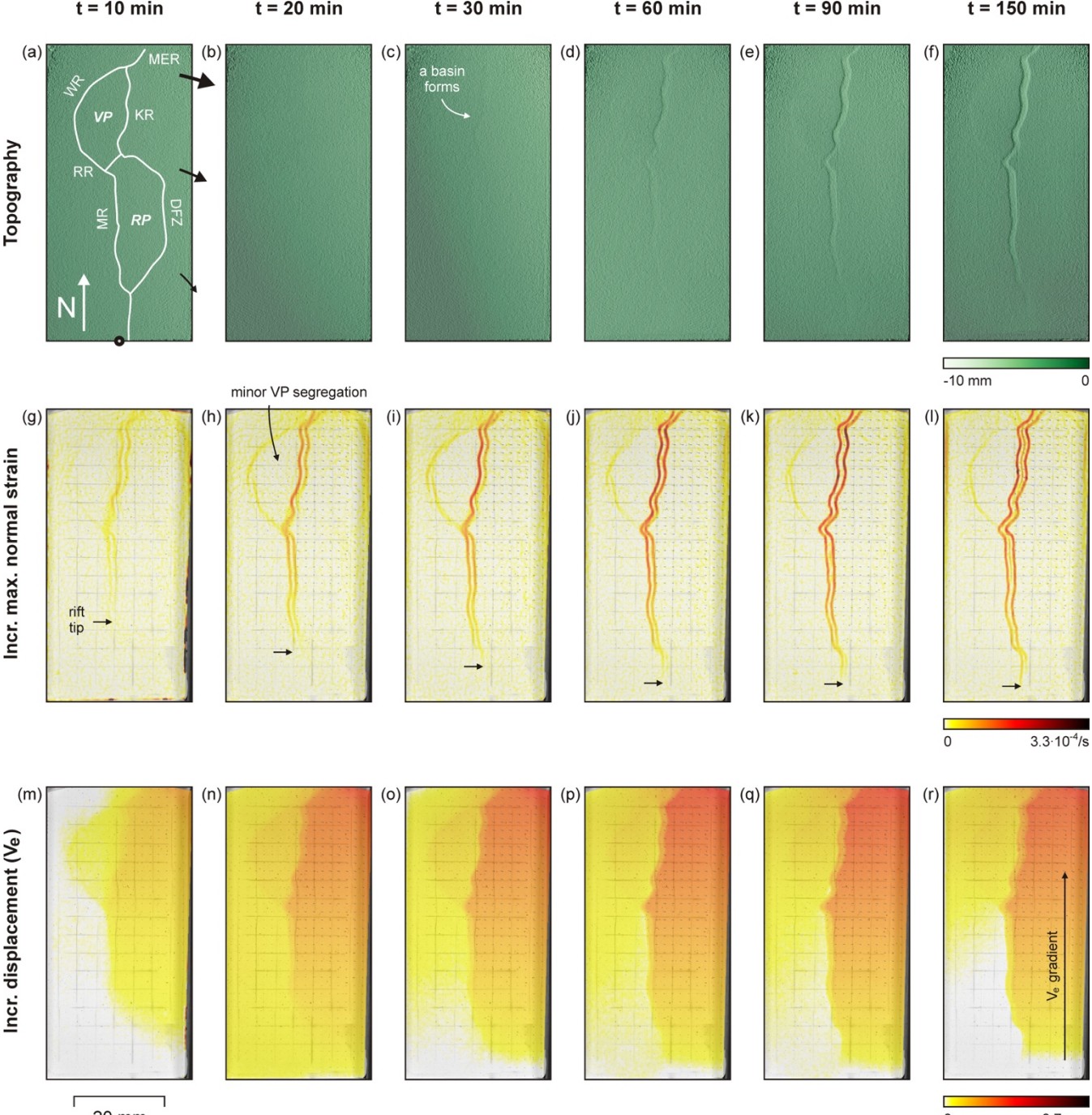

Figure 3: Results of Model A, shown in map view. (a-f) Topography evolution. The initial seed geometry is indicated in panel (a). Lighting direction: from the left. (g-l) Incremental maximum normal strain (m-r). Incremental eastward displacement (Ve). Increments for digital image correlation (DIC) analysis: 10 minutes of divergence. DFZ: Davie Fracture Zone, KR: Kenya Rift, MER: Main Ethiopian Rift, MR: Malawi Rift, RP: Rovuma Plate, RR: Rukwa Rift, VP: Victoria Plate, WR: Western Rift.

## 3.2. Model B

Model B, has the same standard seed thickness as Model B (ø = 4 mm), but the seeds trace less of the present-day EARS plate boundaries (Fig. 4). A major difference with Model A is that in Model B, the simulated Western Rift is much better developed, and there is only a minor connection between the simulated Kenya Rift and the Malawi Rift, whereas Model A features a clear through-going rift basin.

The DIC results provide important additional details (Fig. 4g-r). As in Model A, deformation already localizes in the early stages, before it becomes visible at the model surface. We also observe that most structures are already activated around t = 20 min. The main differences with Model A are related to the increased development of the simulated Western Rift, as highlighted by the eastward displacement analysis (Fig. 4m-r). This analysis shows the same general rotation imposed along the eastern bound of the model, but the development of the simulated Western Rift causes the presence of the Victoria Plate model equivalent, which undergoes an independent counter-clockwise rotation (as indicated by the decreasing eastward displacement towards the north, Fig. 4m-r).

Further important details are similar to Model A in that we observe a relatively fast early southward propagation of active rifting followed by a slow propagation during the rest of the model run (Fig. 3g-l), and intra-rift deformation along the most developed rift basin (the simulated Kenya Rift, Fig. 3l). An additional observation from DIC data is that the simulated Western Rift initiates its development in the south, and gradually propagates northward, eventually deflecting northward (Fig. 4g-l). Finally, both the DIC and topography data show that the link between the simulated Western and Malawi rifts (i.e. the Rukwa Rift model equivalent), which is 45˚ oblique to the model axis, is rather narrow compared to those rift segments that are more parallel to the model axis (Fig. 4g-r).

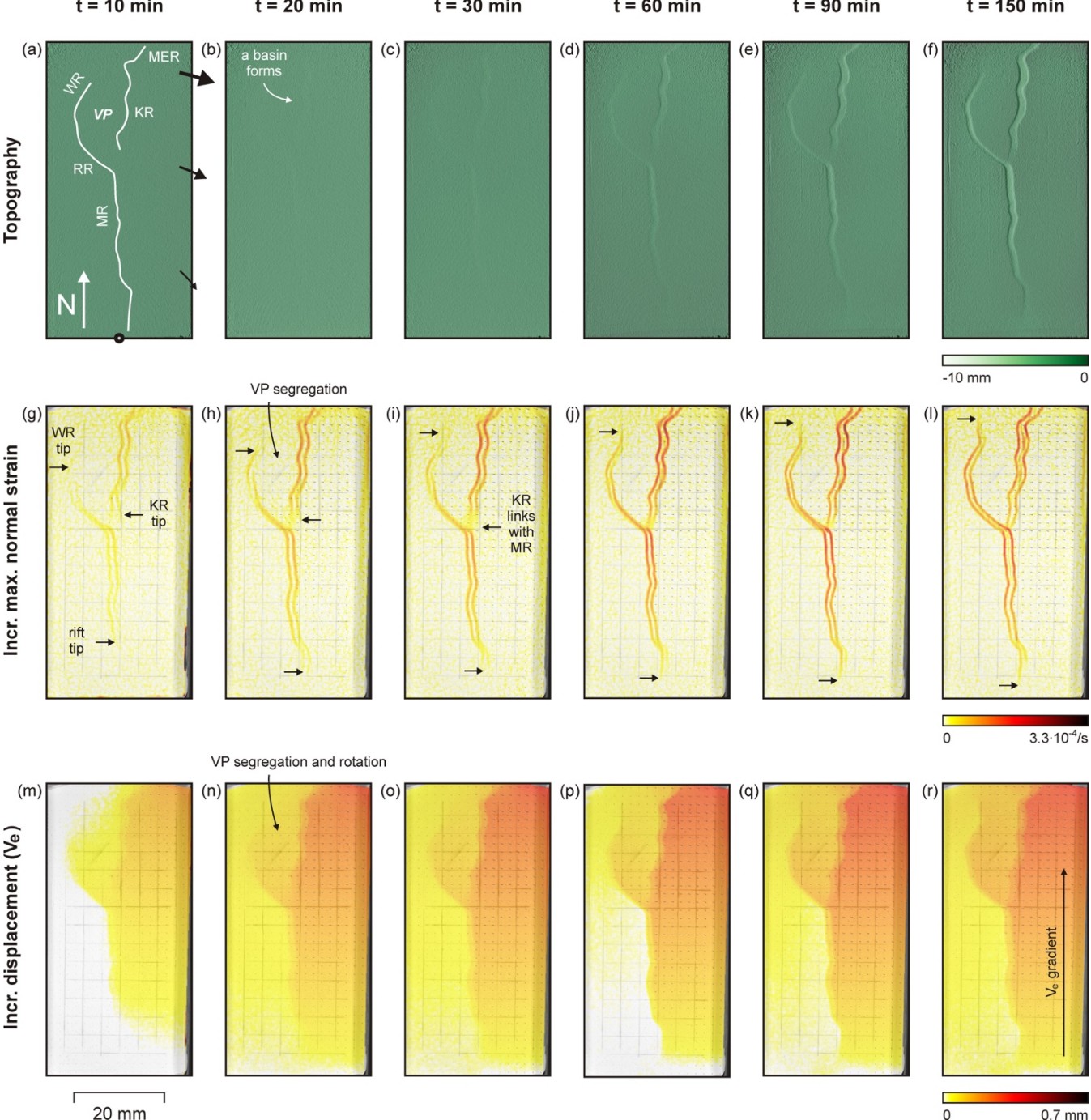

Figure 4: Results of Model B, shown in map view. (a-f) Topography evolution. The initial seed geometry is indicated in panel (a). Lighting direction: from the left. (g-l) Incremental maximum normal strain (m-r). Incremental eastward displacement (Ve). Increments for digital image correlation (DIC) analysis: 10 minutes of divergence. KR: Kenya Rift, MER: Main Ethiopian Rift, MR: Malawi Rift, RR: Rukwa Rift, VP: Victoria Plate, WR: Western Rift.

### 3.3. Model C

Model C is a rerun of Model B, with a slightly reduced seed length in both the simulated Kenya and Western Rifts, so that the seeds do not laterally overlap (similar to the models from Zwaan et al. 2020, Figs. 1d, 5a). This reduced seed length in Model C presence reduces the topographical impact of the simulated Kenya and Western Rifts, even though DIC results show (distributed) extension past the rift tips (Fig. 5f, l). Note that the simulated Western Rift does not deflect as far northward as in Model B, but does generally propagate northward along the trace of the seed, and as a result of its activity the counter-clockwise rotating Victoria model equivalent is established (Fig. 5g-r). Parallel to its Western Rift model equivalent, the simulated Kenya Rift propagates southward, but the southern rift tip ends in a diffuse deformation zone (Fig. 5g-l). Similar to Models A and B, overall southward propagation of extension is observed, decelerating after 20 minutes but never fully halting (Fig. 5g-l), and intra-rift deformation is localizing along the axes of the most developed rift segments (Kenya and Malawi Rifts, Fig. 5l). Furthermore, as seen in Model B, the ca. 45° oblique (NW-SE) simulated Rukwa Rift is rather narrow compared to other, less obliquely oriented, rift segments (Fig. 5f, l).

### 3.4. Model D

Model D is a rerun of Model A, with a double thickness seed (ø = 8 mm) tracing the Western Rift (Figs. 2e, 6). In contrast to Model A, however, the dominant rift branch in the north of the model is the simulated Western Rift (instead of the Kenya Rift) (Fig 6f, l). Hence, we obtain a largely westwardly curved rift system in the north of the model, with only minor deformation at the simulated Kenya Rift (Fig. 6f, l), and a very faint reactivation of the eastern borders of simulated the Rovuma Plate since initiation of the model run (Fig. 6g-l). As a result of this configuration, all domains east of the dominant Western Rift model equivalent are moving eastward, where the simulated Kenya Rift slightly mitigates this effect so that a generally eastward moving Victoria Plate model equivalent can be observed, moving a bit slower than the easternmost domain (Fig. 6m-r). Note that in contrast to Models B and C, this microplate does not rotate as indicated by the lack of a clear north-south displacement gradient (Figs. 4m-r, 5m-r, 6m-r). Also the Rovuma plate model equivalent exhibits a faintly different eastward motion compared to the general easternmost domain (Fig. 6m-o).

Also in this model, we see that the general southward propagation of extension is strongly decelerating after 20 min, but no northward propagation is seen in the simulated Western Rift (Fig. 6g-l). Furthermore, not only is the oblique Rukwa Rift segment between the simulated Western Rift and Malawi Rift narrower in shape than the ca. N-S oriented rift basins, this is also true of the highly oblique link between the simulated Western Rift and Main Ethiopian Rift in the north (Fig. 6f, l). Finally, although it is not that clearly visible in the images, there are some hints of intra-rift deformation in the simulated Malawi Rift, Western Rift, and Main Ethiopian Rift (Fig. 6l).

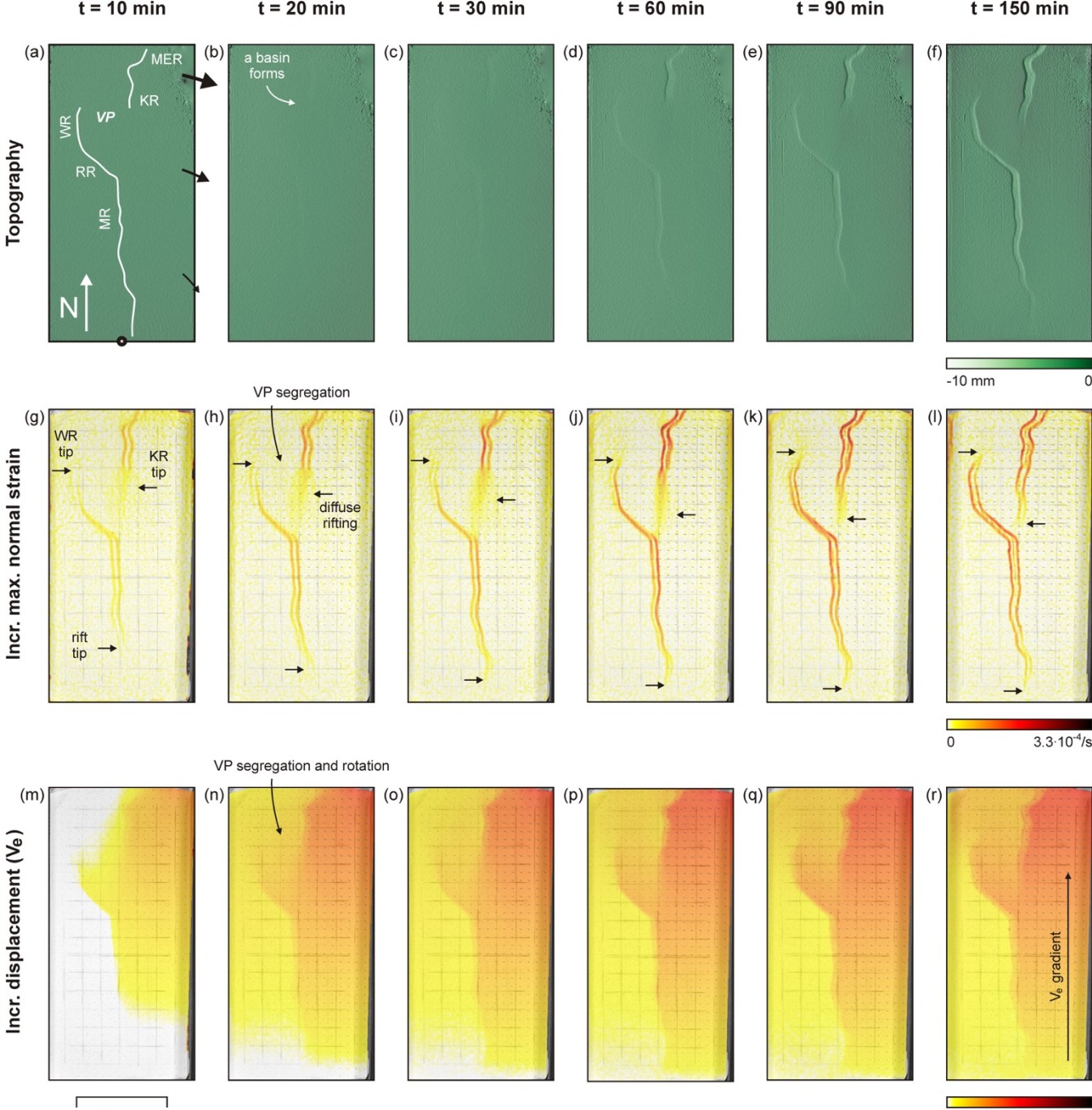

**Figure 5: Results of Model C, shown in map view. (a-f) Topography evolution. The initial seed geometry is indicated in panel (a). Lighting direction: from the left. (g-l) Incremental maximum normal strain (m-r). Incremental eastward displacement (Ve). Increments for digital image correlation (DIC) analysis: 10 minutes of divergence. KR: Kenya Rift, MER: Main Ethiopian Rift, MR: Malawi Rift, WR: Western Rift.**

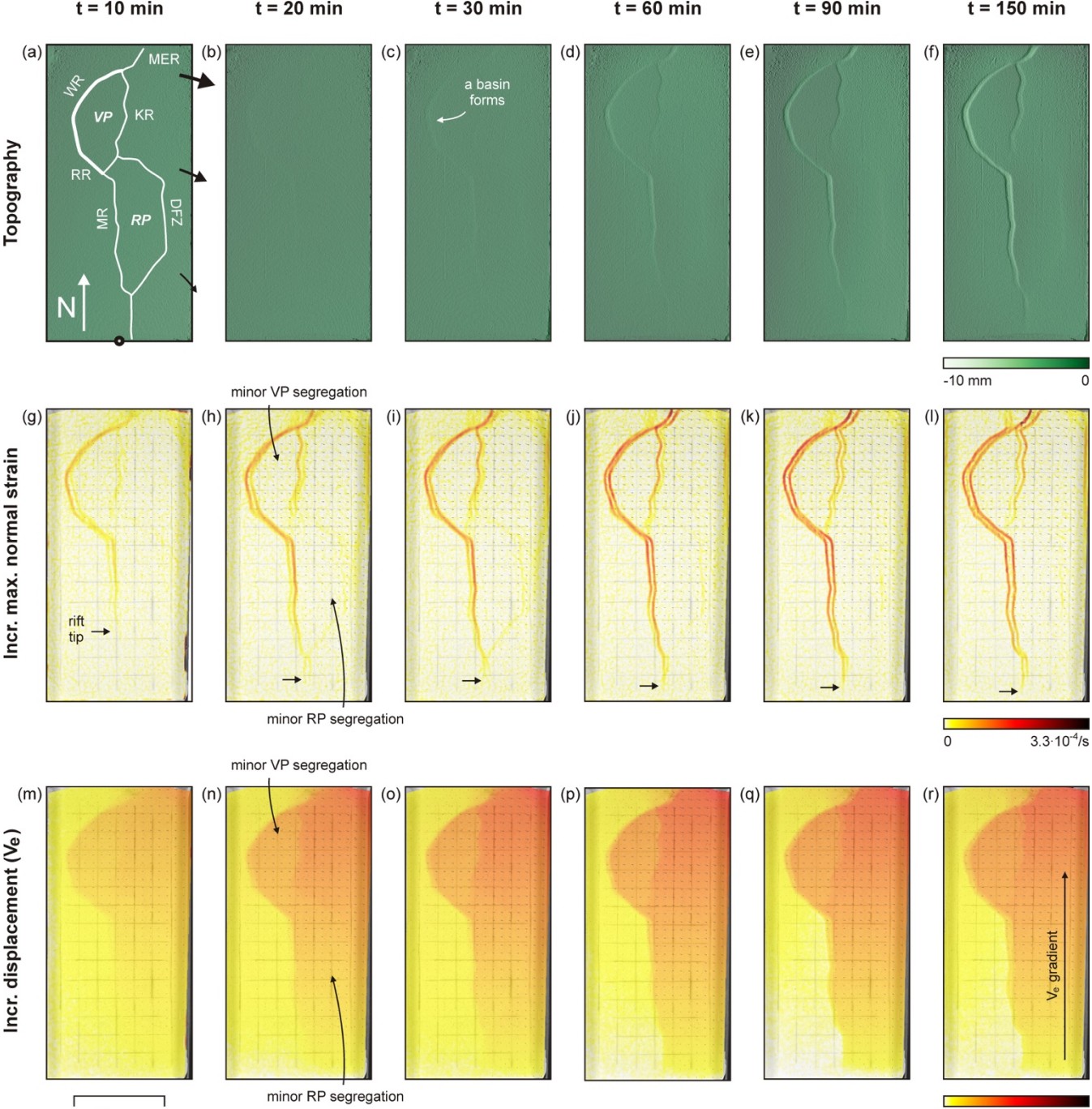

**Figure 6: Results of Model D, shown in map view. (a-f) Topography evolution. The initial seed geometry is indicated in panel (a), where the Western Rift seed had a double diameter (ø 8 mm). Lighting direction: from the left. (g-l) Incremental maximum normal strain (m-r). Incremental eastward displacement (Ve). Increments for digital image correlation (DIC) analysis: 10 minutes of divergence. DFZ: Davie Fracture Zone, KR: Kenya Rift, MER: Main Ethiopian Rift, MR: Malawi Rift, RP: Rovuma Plate, RR: Rukwa Rift, VP: Victoria Plate, WR: Western Rift.**

### 3.5. Model E

In Model E, we apply a standard thickness seed that follows the westernmost limits of the EARS (Fig. 7a). The results are very similar to those of Model D, even though the seed geometry is quite different (Figs. 6, 7). We observe the development of a
rift system along the length of the seed, following the various curves of said seed. After 20/30 minutes the earliest structures become visible at the surface (Fig. 7b, c), whereas DIC analysis shows early extension along most of the seed, which propagates southward until the 20-minute mark, before slowing down drastically (Fig. 7g-l). Similar to the structures in Model D, the oblique parts of the rift system are narrower, and all domains east of the rift system move eastward as shown by the north-south displacement gradient (Fig. 7f, l, m-r). Furthermore, the DIC results also show some localization of extension along the
axis of the model during the first 30 minutes of model development (Fig. 7g-i).

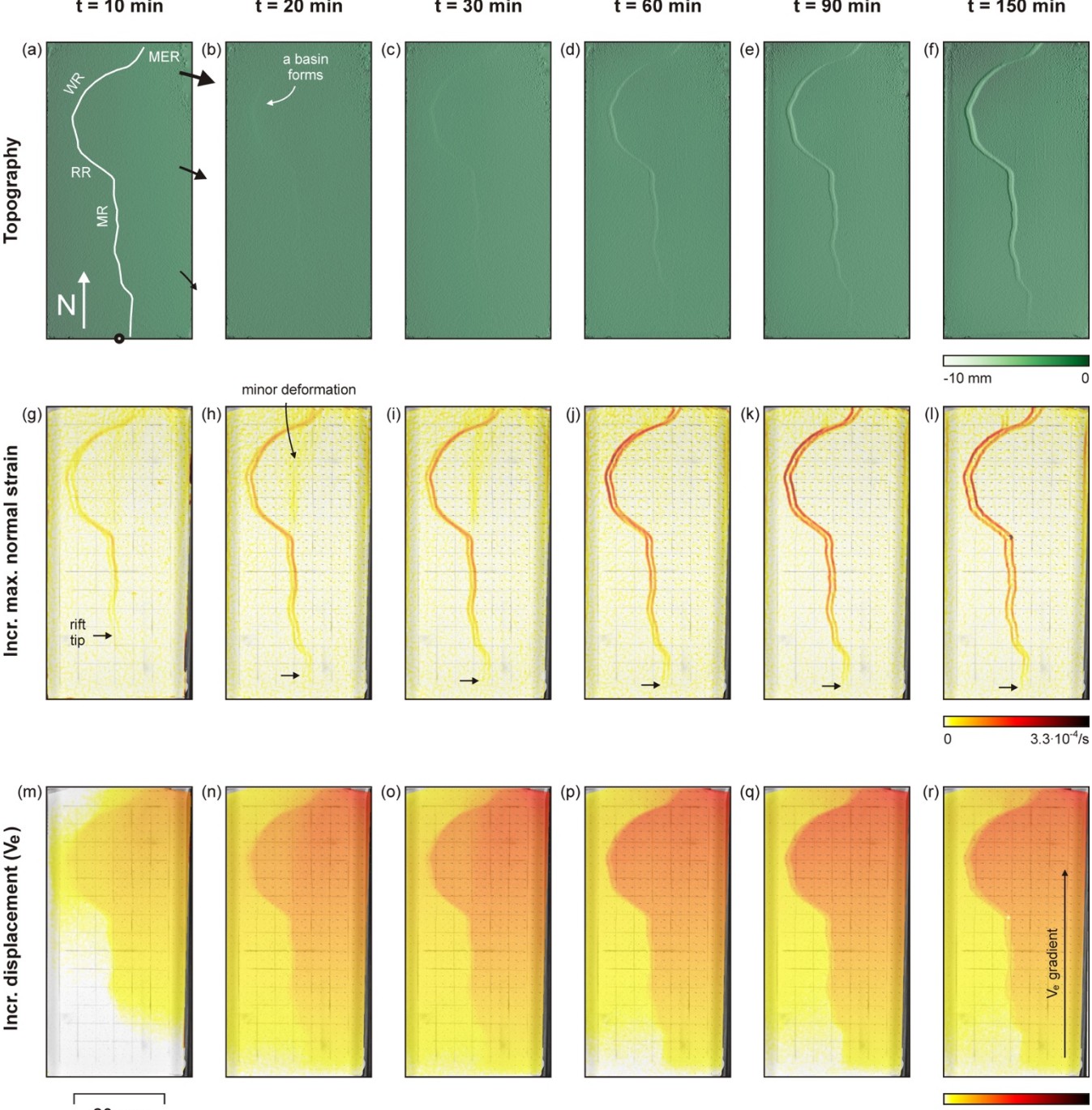

**Figure 7: Results of Model E, shown in map view. (a-f) Topography evolution. The initial seed geometry is indicated in panel (a).**
**Lighting direction: from the left. (g-l) Incremental maximum normal strain (m-r). Incremental eastward displacement (Ve). Increments for digital image correlation (DIC) analysis: 10 minutes of divergence. MER: Main Ethiopian Rift, MR: Malawi Rift, RR: Rukwa Rift, WR: Western Rift.**

## 4. Discussion

### 4.1. Synopsis and comparison to previous modelling studies

In Figure 8 we present a summary overview of our model results, taken at the end of each model run, which forms the basis for a synopsis of model results and comparison to previous modelling studies presented in the following sections.

#### 4.1.1. Rotational rifting

The first important insight from our model results is that the rotational divergence boundary condition leads to an overall northwardly increase in divergence velocity and thus extension that correlates with the most developed rift basins occurring in 345 the northern part of the models (Figs. 3-8). This correlation has been observed in many previous modelling studies involving rotational divergence (e.g., Souriot and Brun 1992; Benes and Scott 1996; Sun et al. 2009; Molnar et al. 2017, 2018; Mondy et al. 2018; Zwaan and Schreurs 2020; Khalil et al. 2020; Zwaan et al. 2020; Schmid et al. 2022a, b).

A further observation related to rotational divergence in our models is the initial fast rift propagation of extension towards the 350 rotation axis in the south, followed by a deceleration and a slow propagation in the later stages of the model run. This effect was previously reported by Schmid et al. (2022a, b), and is upon closer inspection also visible in the models by Mondy et al. (2018), Maestrelli et al. (2020) and Zwaan et al. (2020). Indeed, this fast propagation of extension is not directly obvious from topography data and requires DIC results to be revealed, highlighting the need for such DIC analysis to monitor deformation in analogue models. It also shows that rift propagation in the models is in fact a much faster process than perhaps previously 355 interpreted from topography data: very early on in the evolution of the models, rifting has started along most of the rift system, establishing the template for its subsequent structural evolution.

#### 4.1.2. Seed geometry and microplates

Rotational divergence itself would be expected to produce a linear rift system along the axis of the model (e.g. Martin 1984; Benes and Scott 1996; Mondy et al. 2018; Zwaan et al. 2020; Schmid et al. 2022a, b), but the structural inheritance applied in 360 our models forces the model to deviate from this template, generating a range of rift basin arrangements. When comparing the various seed geometries we applied to the final structures produced in the models, the relations are somewhat complex (Fig. 8). In Model A, representing the end member with the dominant rift structures situated farthest to the east, not all seeds are reactivated (Figs. 3, 8a-d). It seems that extension preferentially localized in the north of the model, along the simulated Main Ethiopian and Kenya Rift where the largest degree of extension is applied, before swiftly propagating south along the strike of 365 the model, mostly avoiding the seeds tracing the Western Rift and the eastern margin of the Rovuma Plate (Fig. 3g-l). We also found some localization of deformation along the model axis in Model E (Fig. 7g-i). It may very well be that this orientation of rifting (i.e. near the axis of the system) is ideal in this type of model, so that the seeds that are thus aligned are preferentially activated at the expense of those that are off-centre, in this case the Western Rift and eastern margin of the Rovuma Plate.

Such preferential reactivation of more favourably oriented weaknesses has been observed in various previous modelling studies (e.g. Henza et al. 2010, 2011; Zwaan and Schreurs 2017; Molnar et al. 2019, 2020; Maestrelli et al. 2020; Wang et al. 2021; Zwaan et al. 2021a, 2022a; Bonini et al. 2023).

However, when these seeds are not fully linked up as in Models B and C, deformation is forced to follow the offset seeds, resulting in a rift pass or overlap configuration (Nelson et al., 1992; Hieronymus, 2004; Kolawole et al., 2021, Figs. 4, 5, 8e-l). The overlapping configuration of these rifts, where one rift becomes dominant over the other along strike, causes the rotation of the central rift pass area (in this case the simulated Victoria Plate, Figs. 4m-r, 5m-r, 8h, l). Such rotations of rift pass blocks have been reported in numerous modelling studies (e.g. Oldenburg and Brune, 1975; Katz et al., 2005; Tentler and Acocella, 2010; Zwaan et al., 2016, 2018, 2020; Brune et al., 2017; Molnar et al., 2017, 2018; Glerum et al., 2019; Neuharth et al., 2021). Important in these systems is that both overlapping rifts propagate as the block rotates, which is also seen in our Models B and C (Figs. 4m-r, 5m-r,). In combination with regional-scale rotation, this means that active extension (i.e. rift basins) can propagate both (regionally) towards and (locally) away from the rotation pole, as proposed by Zwaan and Schreurs (2020) (Figs. 4g-l, 5g-l).

Model D (Fig. 8m-p) continues the westward shift in rift localization shown in Models B and C (Fig. 8e-l). This more westward concentration of rifting is due to the thicker seed tracing the Western Rift, which overrides the tendency of the model to preferentially localize deformation along the central model axis that is shown in Models A and E (Figs. 7g-i, 8). A thicker seed represents a more impactful weakness in the system, and such weaknesses naturally attract more deformation, as also observed in previous modelling studies (e.g. Henza et al., 2010, 2011; Wang et al.., 2020; Zwaan et al., 2021a, Osagiede et al., 2021) Even so, the Kenya Rift seed still localized part of the deformation in the system, creating a Victoria Plate model equivalent as in Models B and C, but with the crucial difference that this plate does not rotate counter-clockwise (Figs. 6m-r, 8p). Instead, in Model D it merely moves along with the rotation of the eastern domain, albeit at a slower rate due to the presence of the simulated Kenya Rift (Fig. 6m-r). Similar to Model A, the eastern part of the Rovuma Plate did not reactivate in Model D, very likely due to the same unfavourable arrangement with respect to the overall tectonic set-up (Figs. 6, 8a-d, 8m-p).

Model E is the other end-member model, where the single seed localizes deformation along the Western Rift and Malawi Rift traces (Fig. 8q-t). Even though no seed was present along the Kenya Rift trace, there are some hints of minor early extension localization there, highlighting the model's preference to localize deformation along its central axis (Fig. 6g-j). Nevertheless, the experiment results in a solid eastern domain that rotates eastward in a clockwise orientation (Figs. 6g-l, 8t), without any hint of the Victoria microplate.

### 4.1.3. Additional observations: intra-rift deformation and oblique rifts

In addition to the large-scale observations described above, some more detailed insights can be gained from our models. Firstly, the DIC analysis shows the initial localization of boundary faults flanking the rift segments, followed by intra-rift deformation along the central axis of the most developed rift basins (Fig. 8c, g, k, o, s). This shift of deformation has been observed on DIC results of previous models (Schmid et al. 2022a, b) as well, and can be explained by the rise of the viscous layer below the rift axis in this type of brittle-viscous models (e.g., Zwaan et al. 2018; 2016; 2020), while the rift boundary faults have accommodated most of their slip at that point in time.

The various orientations of the rift segments provide some final insights. Indeed, there where the rift basins are oriented obliquely with respect to the model axis (and to the general extension direction), they are much narrower than those rift basins that are parallel to the model axis (Fig. 8b, f, j, n, r). This is a straightforward effect of the local oblique extension direction: the larger the extension obliquity, the more the system moves towards steep strike-slip faulting (e.g. Corti et al. 2007; Zwaan et al. 2016). Here it must be noted that for an exact determination of the extension obliquity, the rotational divergence and the curved displacement traces needs to be taken into account (Fig. 8a, e, i, m, q), but the general orientation of the rifts with respect to the model axis provide a first order indication that is sufficiently accurate (see also the Appendix).

A detail that is however not captured in our models are offset en echelon structures, which are typical of oblique extension models (e.g., Withjack and Jamison 1986; Tron and Brun 1991; Bonini et al. 1997; Keep and McClay 1997; Zwaan et al. 2021a, 2022a). Especially the simulated Western Rift should be expected to contain various offset sub-basins, as shown by previous models (which do however not incorporate rift propagation, Corti et al. 2007). This is likely due to the large scale of our model set-up, as well as the general brittle-viscous set-up with seeds preventing the development of such details (compare our results with Figs. 5 and 7 in Zwaan and Schreurs 2017). As such, our models are best suited to reveal the large-scale trends in such settings.

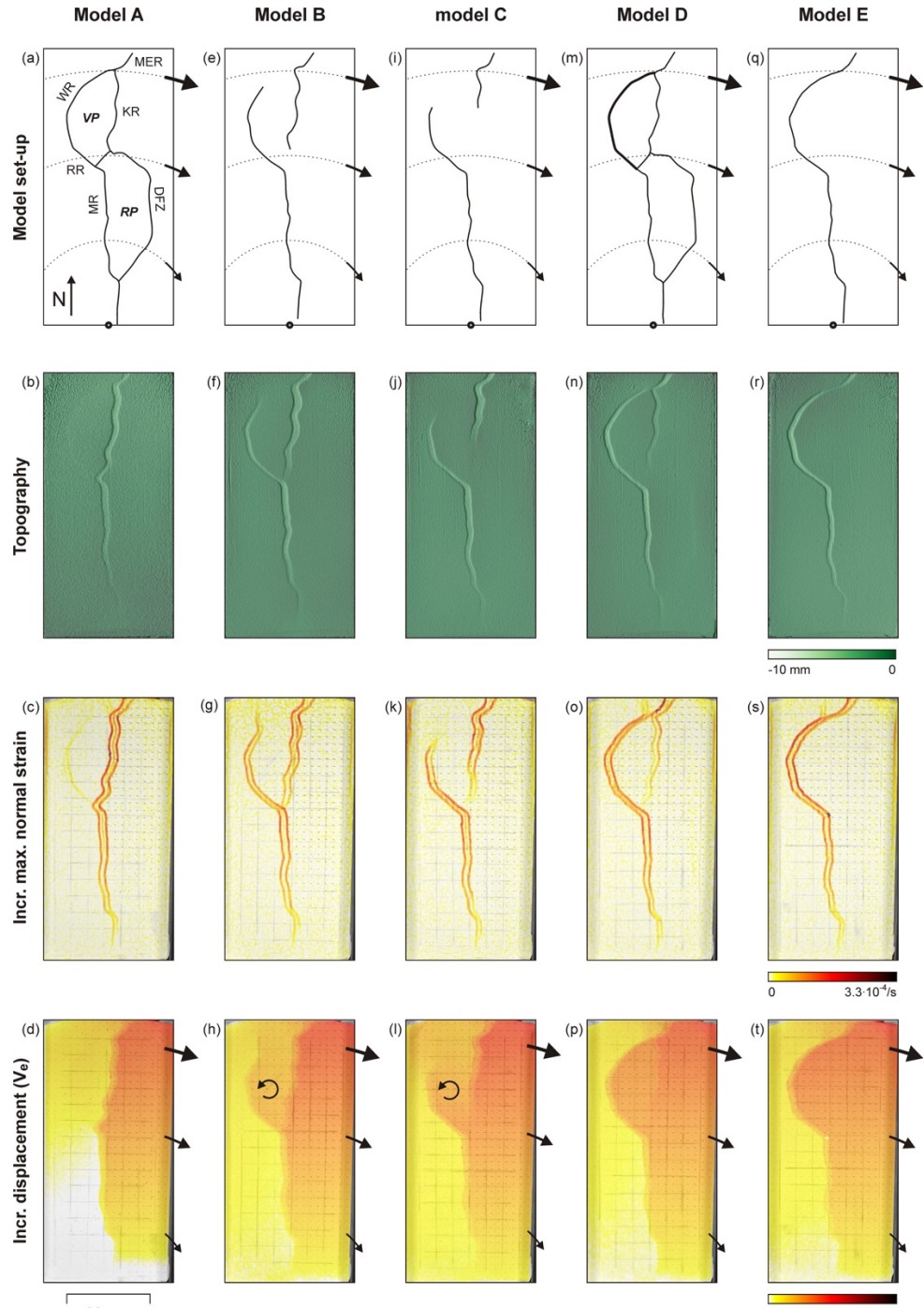

**Figure 8: Overview of final stage (t = 150 min) model results from Models A-E, shown in map view. (a-e) General model set-up detailing seed geometries and model kinematics. (f-j) Final model topography. (k-o) Incremental maximum normal strain. (p-t) Incremental eastward displacement (Ve). Increments for digital image correlation (DIC) analysis: 10 minutes of divergence. DFZ: Davie Fracture Zone, KR: Kenya Rift, MER: Main Ethiopian Rift, MR: Malawi Rift, RR: Rukwa Rift, RP: Rovuma Plate, VP:**
**Victoria Plate, WR: Western Rift.**

## 4.2. General model limitations

Before comparing our model results with the EARS, it is important to point out some general limitations of our models. Firstly, as mentioned in section 4.1.3, our models should be used for interpreting the evolution of large-scale tectonic structures as we apply a highly simplified, brittle-dominated lithosphere. Furthermore, the modelling of rotational divergence on a sphere (as

is the case in nature) is done on a flat plane (Zwaan et al. 2020), which may cause some distortions towards the rotation pole and affect the rate of rift propagation in our models (Schmid et al. 2022a, b). Yet these minor effects are not considered to significantly undermine our model results. Some boundary effects occur along the long edges of our model, as shown by the slight rotation of the westernmost domain in our models, but also the impact of these boundary effects on the large-scale evolution of the model are minor. Another limitation is the lack of surface processes (erosion and sedimentation) in our models,

which are known to affect rift evolution (e.g., Burov and Cloetingh, 1997; Buiter, 2009; Neuharth et al., 2022). However, the analogue models by Zwaan et al. (2018a) suggest that surface processes do not significantly alter the large-scale structures of young continental rift systems, and as such, our model results remain applicable on a first-order scale. Finally, we apply far-field tectonic rotation of the Somalian Plate in our models with deformation driven by sidewall motion (Fig. 2), thus ignoring the gravitational energy potential or mantle flow that may be largely driving EARS development in nature (e.g., Bagley &

Nyblade, 2013; Kendall and Lithgow-Bertelloni, 2016; Rajaonarison et al., 2021, 2023). Nevertheless, our models still provide valuable insights into the links between Somalian Plate rotation and the evolution of the EARS, even if this rotation would be a consequence rather than the primary driver of EARS development.

## 4.3. Comparison to the EARS

### 4.3.1 Southward rift propagation along the EARS

Of the five rifting models completed for this study, Model C provides the best fit with the present-day tectonic setting in the EARS (Fig. 9a, b), and we can use the model to discuss the evolution of the natural example. First of all, we can infer a general southward propagation of rift basin development along the whole length of the EARS, as previously proposed by Zwaan and Schreurs, 2020), due to the rigid clockwise rotation of the Somalian Plate (e.g., Saria et al., 2014, Stamps 2021, Fig. 1a). This

southward propagation is in line with data from the EARS suggesting a southward younging direction of rifting (Chorowicz, 2005; Macgregor, 2015). Importantly, our models also suggest that deformation may already have rapidly propagated over large parts of the system during the earlier phases of EARS development, which is a logical result of rigid plate rotation causing rotational deformation along the whole length of the EARS. Such a fast southward propagation of EARS structures over the last 20 Myr is inferred by Macgregor (2015) and Purcell (2018) (Fig. 1c-e).

We must however point out that the exact age of the various rift basins along the EARS is debated (Purcell, 2018; Zwaan and Schreurs, 2020, Michon et al., 2022, Martin, 2023, and references therein), and such early extension may have occurred without

leaving behind clear field evidence due to the initially limited amount of extension. Moreover, studies have shown variations in lithospheric thickness and strength along the EARS (e.g., Globig et al., 2016; Daly et al., 2020), which can have halted or promoted rift development (e.g., Benes & Scott, 1996; Brune et al. 2017, Fig. 1b). This influence of lithospheric thickness is to a degree also illustrated by the delayed southward propagation of the simulated Kenya Rift when a shorter seed is included in our models (compare Models B and C in Fig. 9). A further complication in this context is the magma-rich nature of the EARS, which may cause magmatic underplating or strongly localize rift deformation (magma-assisted rifting, Buck, 2004, 2006), so that rift basins may develop faster than in magma-poor settings, skewing the tectonic signal. Such magmatism also provides concrete time constraints on rift initiation (e.g. Ayalew et al., 2006; Wolfenden et al., 2005; Michon et al., 2020), but these records may not always be fully preserved in the sub-aerial conditions of the EARS, as is the case for any syn-rift sediments as well. It is thus clear that continued research is required to improve our understanding of the large-scale evolution of the EARS. Still, our model results suggest that rigid rotational motion of the Somalian Plate rotation can readily explain the general southward propagation of rifting as proposed by Chorowicz, (2005), and shown on maps by Macgregor (2015) and Purcell (2018) (Fig. 9).

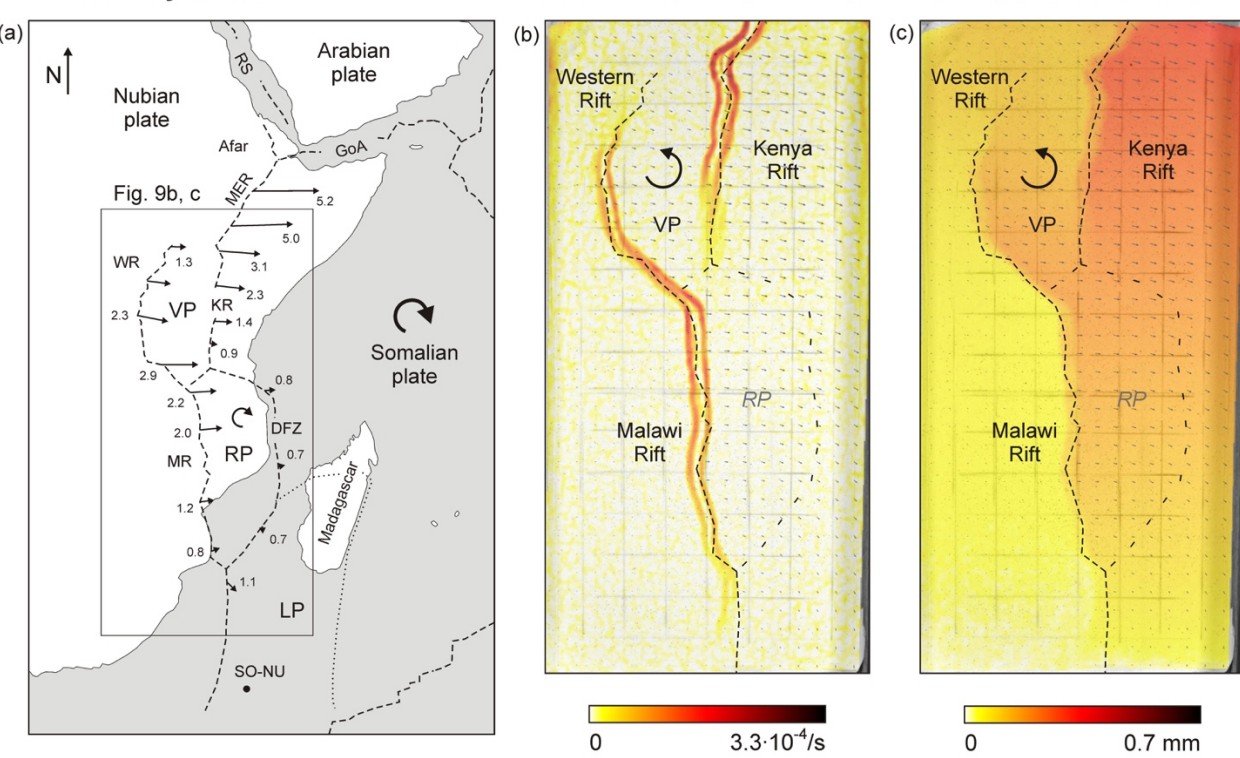

**Present-day EARS**

(a)

**Incr. MSN** (Model C, t = 150 min)

(b)

**Incr. Vₑ** (Model C, t = 150 min)

(c)

0     $3.3 \cdot 10^{-4}$/s

0     0.7 mm

## Proposed evolution of the East African Rift System

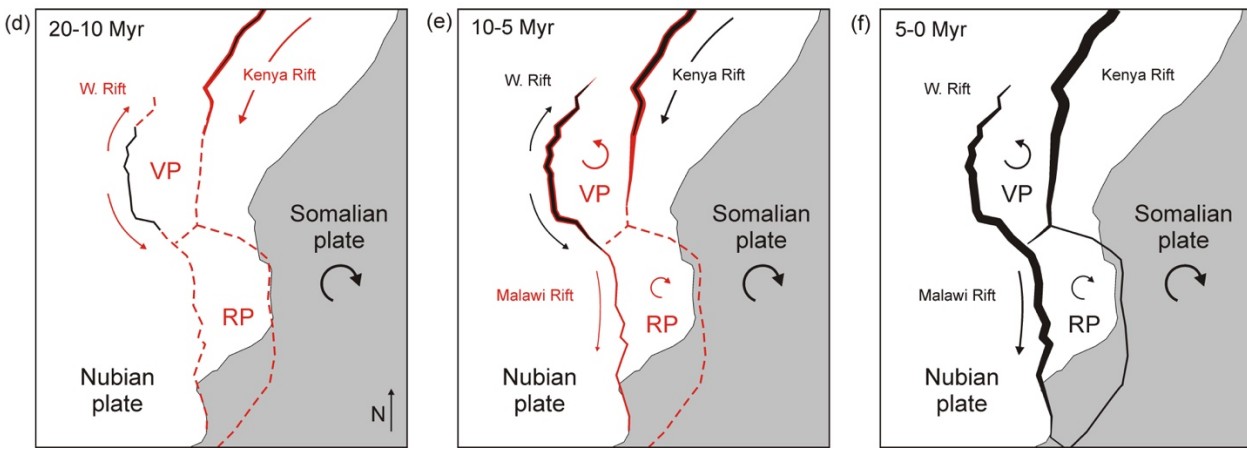

(d) 20-10 Myr

(e) 10-5 Myr

(f) 5-0 Myr

**Figure 9: Application of our model results. (a-c) Comparison between the East African Rift System (EARS) and our Model C results:**
**(a) tectonic map of the EARS, modified after Saria et al. (2014). (b) Incremental maximum normal strain localization, which follows**
**the general trace of the EARS, and (c) incremental eastward displacement (Ve), which mimics the large-scale plate divergence**
**pattern of the EARS, at the end of the model run. (d-f) Proposed evolution of the EARS based on Model C (and D), involving rapid**
**southward propagation of rift-related extension (red), followed by development of discrete rift basins (black). Image modified after**
**Macgregor et al. (2015), Purcell (2018) and Zwaan and Schreurs (2020). DFZ: Davie Fracture Zone, GoA: Gulf of Aden, KR: Kenya**
**Rift, LP: Lwandle Plate, MER: Main Ethiopian Rift, MR: Malawi Rift, RP: Rovuma Plate, RS: Red Sea, SO-NU: Somalia Nubia**
**rotation pole, WR: Western Rift.**

### 4.3.2. Microplates within the EARS

A further key observation in our Model C concerns the evolution of microplates. The simulated Victoria Plate segregated and started rotating in a counter-clockwise early on during the model run, as a necessary result of the rapid nucleation of the overlapping Western Rift and Kenya Rift model equivalents (Figs. 5m-r, 8l). The same rift pass mechanism is indeed clearly active in the EARS, (Glerum et al., 2020, Zwaan and Schreurs, 2020), where GPS observations demonstrate the on-going rotation of the Victoria Plate (Saria et al., 2014; Stamps et al. 2021) (Figs. 1a, 9). From our Model C results, we can propose the Victoria Plate to have been present and rotating ever since the development of the Western and Kenya Rift started around 12 Ma (Fig. 9d), and should have been in full swing by 5 Ma when both rifts were well-established (Fig. 9e) (Macgregor, 2015). Similar to our models, we should then also expect the Western Rift to have propagated northward since its early inception, and the Kenya rift should have propagated southward (all while the rift system as a whole was propagating southward, see section 4.3.1), which is in general agreement with observations from the EARS (Macgregor, 2015; Purcell, 2018; Martin, 2023). During (the early stages of) the Model C run, the southern tip of the simulated Kenya rift also exhibits the diffuse style of deformation observed in nature (Figs. 1b, 5g-l).

In contrast to the Victoria Plate, the Rovuma Plate is not clearly represented in our Model C (although the Rovuma Plate is slightly developed in Model D, Figs. 6, 8). The Rovuma Plate is situated close to the rotation pole and does involve very minor (<1 mm/yr), but constant motion along its eastern border (Saria et al., 2014; Stamps et al. 2021, Fig. 1a) and as such, its rotation is controlled by the divergence gradient along the Malawi Rift. Yet, a complete model of the EARS will need to incorporate the evolution of the Rovuma Plate as well, since it does take up about 30% of the plate divergence in the southern part of the EARS (Saria et al. 2014; Stamps et al. 2021, Fig. 1a). Similarly, the limited motion of the elusive Lwandle plate SE of the Rovuma plate, which is not included in this study due to practical limitations, needs to be addressed too (Fig. 1a). In fact, it appears that there is no consensus regarding the actual extent of the EARS (Michon et al., 2022), and there may be other rift branches and microplates that may have to be considered (Daly et al., 2020; Martin, 2023; Stamps et al., 2021). What can however be predicted from our models is that the Rovuma Plate, similar to the Victoria plate, may have been segregated at an early stage of EARS development as extension rapidly propagated southward (Fig. 9e). We refrain from making such predictions regarding the Lwandle Plate, given its position near the present-day Nubia-Somalia rotation pole (Fig. 1a).

### 4.3.3. Intra-rift deformation and oblique rifts in the EARS

Two further details observed in our models are also relevant to our interpretation of the EARS. The first is the localization of intra-rift extension along the central axis of the most evolved rift basins (Fig. 8c, g, k, o, s). Such localization is also observed in the MER, which represents the most evolved part of the EARS, except for the Afar triangle (Fig. 1a). Here, the so-called magmatic segments belonging to the Wonji Fault Belt are situated along the rift axis, and are found to accommodate a significant portion of the total extension in the system as it is nearing break-up (e.g., Ebinger, 2005; Pizzi et al., 2006; Ebinger et al., 2008). It must however be pointed out that our models do not directly simulate the magmatic processes that may promote weakening of the crust in the MER (Buck 2004, 2006). Yet, basin-inward migration is a well-known phenomenon in other (magma-poor) rift basins as well (e.g., Cowie et al., 2005; Sutra et al., 2013). The second observation concerns the impact of rift orientation on rift structures. Although our models do not provide detailed insights into fault structures, the general oblique / strike-slip nature of the obliquely oriented rift segments such as the Rukwa Rift is clearly recognized (Morley, 2010; Glerum et al., 2020, Figs. 8d-t, 9b).

### 5. Conclusion

In this study we present a new series of analogue rotational rifting models specifically tailored to explore the dynamic evolution of the East African Rift System (EARS) and its link with the rotation of the Somalian Plate. Our model results lead us to the following conclusions:

- Rotational rifting leads to rift propagation in all our model runs. Yet we need to distinguish between the propagation of rift-related deformation, which can move very rapidly towards the rotation axis, and the surface expression of this rapidly propagating deformation in the shape of discrete rift basins, which can significantly lag behind. We speculate that such rapid propagation towards the Nubia-Somalia rotation pole to the south may have taken place in the EARS as well.

- The different structural weakness geometries we tested in our models lead to a variety of rift system arrangements, of which our model C provides the best fit with the EARS. It is clear from our models that laterally overlapping weaknesses are required to form parallel rift basins and the creation of rift pass structures, possibly leading to the segregation of microplates. These plates can start rotating if the rift basins on both sides are sufficiently developed, as is the case for the Victoria Plate in the EARS. Importantly, this involves the northward propagation of the Western Rift, which goes against the general southward propagation direction of the EARS.

- Additional model observations involve the development of early pairs of rift-bounding faults flanking the rift basins, followed by the localization of deformation along the axes of the most developed parts of the rift system. Such a shift of deformation towards the centre of the rift has been observed along the Main Ethiopian Rift, where it is associated with incipient (magma-rich) continental break-up, but also occurs in many other rift basins around the globe.

- Finally, we observe how the orientation of rift segments with respect to the regional (rotational) plate divergence affect deformation along these segments. Compared to rift segments that are oriented (near-)perpendicular to the plate divergence direction, obliquely oriented rift segments are less wide and show a component of strike-slip deformation (i.e., oblique extension).

- Overall, we find that our model results generally fit the present-day large-scale features and ongoing deformation along the EARS, and have implications for general rift development, rift basin propagation, and for the segregation and rotation of microplates.

## Author contribution

FZ designed and carried out the experiments, analysed the experimental results, and wrote the original draft of this paper. GS secured funding, took care of project administration and critically reviewed the original draft.

## Competing interests

The authors declare that they have no competing interests.

## Data availability

Extensive overviews and videos of our modelling results are provided in the form of a GFZ data publication (Zwaan et al. 2023). NB: this data publication is in preparation but the files can already be accessed in this Dropbox folder: https://www.dropbox.com/sh/6k34psul6cij4dh/AABpt-cnM1OyIEdolfTszmmya?dl=0

## Acknowledgements

We thank Timothy Schmid from the University of Bern Tectonic Modelling group for helping with the DIC analysis, and to Florian Ott and Kirsten Elger from GFZ Potsdam for helping us prepare the GFZ data publication containing the supplementary data of this paper (Zwaan et al. 2023). This project was funded by the Swiss National Science Foundation (SNSF) through grant 200021-178731 "4D Analogue modelling of oblique rifts and obliquely rifted margins" (https://data.snf.ch/grants/grant/178731). The SNSF also covered the Open Access publication costs. FZ was further supported by a GFZ Discovery Fund fellowship. We thank Cynthia Ebinger and Georgios-Pavlos Farangitakis for their constructive feedback, as well as editor Jordan Phethean for guiding the review process.

## Appendix

The five models presented in the main text all have a minor error in that the location of the rotation axis is in fact somewhat too far to the west (Fig. A1). Although this error does somewhat modify the relative direction of plate divergence along the seeds (Fig. A1a, e), the final result in both Model A, and Model F with the correct rotation axis location are very similar (Fig. A1. Hence, we conclude that the results from our Models A-E as presented in this paper remain valid.

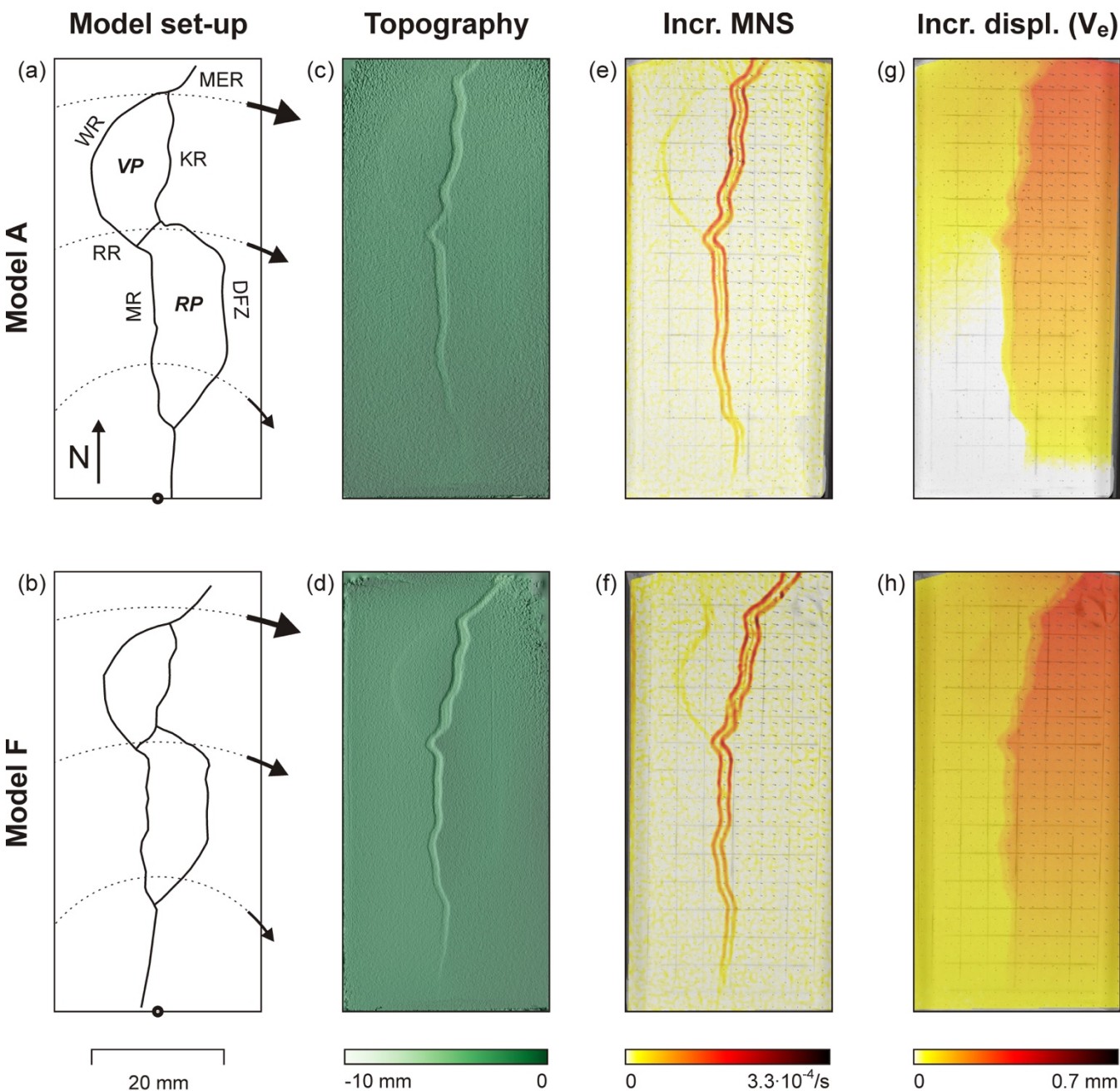

**Figure A1: Comparison of final stage (t = 150 min) model results from Models A and F. (a-b) General model set-up showing seed geometries and model kinematics. (c-d) Final model topography. (e-f) Incremental maximum normal strain (MNS). (g-h) Incremental eastward displacement ($V_e$). Increments for digital image correlation (DIC) analysis: 10 minutes of divergence. DFZ: Davie Fracture Zone, KR: Kenya Rift, MER: Main Ethiopian Rift, MR: Malawi Rift, RP: Rovuma Plate, RR: Rukwa Rift, VP: Victoria Plate, WR: Western Rift.**

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
