# Peer review of "The link between Somalian Plate rotation and the East African Rift System: an analogue modelling study"

_EGUsphere, 2022_

## Author Comment (AC1)

**Reviewer 2 – Georgios-Pavlos Farangitakis**

**Overall assessment:**

The authors present a novel analogue modelling approach through a rotating apparatus to delineate the evolution in response to rotational motions of rifts and use the East African Rift System as a natural example. They explore different rift-seed configurations and how these reflect on various features of the East African Rift System, making observations in the process about rotational stresses, microplate formation and preferential rift (re)activation.

A particularly novel part of this work is that the authors do a direct comparison on the initiation of a rift based on top-view photos, digital topography and particle tracing techniques and discover that rifts cluster far earlier than is generally observed with traditional means of capturing observations.

The only content related comment on this work is that the manuscript is missing a discussion on the potential discrepancies that arise due to a) the non-differentiation of crustal nature within the Somali plate (author models use a uniform continental plate rheology) and b) the absence of a key inheritance feature trending parallel to the extension direction, the Davie Fracture Zone.

The manuscript is well written and concise, and most other comments on this work are formative (consisting of missing scale bars in figures, acronyms, shortening sentences and possibly rephrasing some expressions).

Therefore, I recommend this work to be accepted for publication after these revisions are addressed.

For specific comments see below:

Dr Pavlos Farangitakis, Shell Global Solutions International

- o **Reply:** We kindly thank the reviewer for the positive assessment of our manuscript, and have addressed the comments in the text below.

**General comments:**

- I believe the manuscript would benefit from a discussion on how the difference in the nature of the crust (continental vs oceanic), would affect the evolution of the region. Between Madagascar and Mozambique, there is a stretch of a few 100s of kms of oceanic crust. I understand the limitations of analogue modelling and not being able to place different crustal compositions side by side in such an experiment, however it does merit a discussion point. I have added specific comments on where parts of this could fit through the manuscript

- o **Reply:** The reviewer is right to point out the difference between oceanic and continental lithosphere, and the potential impact of the Davie Fracture Zone. Indeed, the NE margin of the Rovuma plate seems to follow this fracture zone, which we now mention in the text.
  - o However, for the model set-up we consider the whole Somalian plate as a rigid plate. We believe this is permissible since:
    - The configuration of plates southward of the continental East African Rift System remains debated
    - The plate motion velocities south(east) of the (continental) East African Rift System are very minor in comparison to plate motion velocities in the continental East African Rift System
    - These areas are found in close proximity to the Nubia-Somalia rotation pole, which makes their (limited) motion rather complex. Hence we focus our modelling study on the system north of this rotation pole.
    - The Mesozoic rifting that caused the separation of Madagascar from the African continent and the development of the Davie Fracture Zone was not active during the Cenozoic development of the East African Rift System (Phethean et al. 2016).
  - o This is now also addressed in the text.

- Further to the previous comment, part of the key pre-existing structural fabric is missing (such as the Davie Fracture Zone – see Phethean et al. 2016 or Reeves et al. 2016). Why do you consider this to not play a part on the EARS and do not include it in your set-up? It would be useful to see a discussion on what the omission of these two features (oceanic crust and Davie Fracture Zone) entails for your experiments (I suspect not much given that first order similarity is always the key outcome, however, will supplement the manuscript's quality having captured the entirety of the key lithospheric features).

  - o **Reply:** In our models we implemented the main **active** EARS plate boundaries as proposed by e.g. Saria et al. (2014) and Stamps et al. (2021). As such, the Davie Fracture Zone is not completely included (apart from the NE boundary of the Rovuma plate. We have now expanded the discussion of weaknesses in East Africa.

**Specific comments:**

- (L10-15) Short summary: I am not sure whether this is also doubling as a plain language summary or not, if it is the case, consider adding in brackets in the first sentence that the "features splitting continents apart" are called rifts. If not, ignore this comment.

  - o **Reply:** We believe the current text is sufficiently clear (including the use of "Rift" in the name of the East African Rift") and prefer to keep things as they are.

- (L43-56) Introduction: What is missing from this section is a discussion/reference on the relative timing of each of the features on the EARS, particularly if references to

inherited structures are made (such as the ones Glerum et al. 2020 refer to). Might be worth adding also some info on the map the further northward extension of the the Davie Fracture Zone, as you'd expect them to play a part in the plate configuration (See Phethean et al. 2016 this also refers to Fig 1 and later to the set-up).

- o **Reply:** We agree that details on timing is missing here, and have added these in the new text. Timing of East African rift development is now also discussed in more detail in the discussion section.
- o See previous replies regarding the Davie Fracture Zone.

- Figure 1: Can you add the respective scale to each of the sub-panels?

  - o **Reply:** We have added scales to panels (a) and (c)

- (L84-93) Methods: I believe this section would benefit from a small 2-3 sentence section on highlighting the specific novelty of rotational analogue models in general (i.e. the works of Souriot and Brun 1992, Molnar et al. 2017, Farangitakis et al. 2019, 2021)

  - o **Reply:** We understand what the reviewer means, but don't feel that the methods are the right place to do so (we feel it would mostly provide a distraction from our modelling set-up details). But note that we already cited some of these modelling studies in the introduction, which was also rewritten to accommodate suggestions made by reviewer 1 as well.

- (L90-92) Methods: This sentence reads slightly odd. Consider breaking in two or putting the "at a rate of 4mm/h at the farthest point from the rotation axis" in a bracket.

  - o **Reply:** We agree and have made this part of the sentence into a separate sentence following the original sentence.

- (L95-101) Methods: Again, the DFZ is missing from the story here and the oceanic crust as well.

  - o **Reply:** See previous replies to comments on the DFZ and oceanic crust

- (L97-98) Methods: Consider also explaining how these seeds have also been used to simulate inheritance in rifted basins (such as the Gulf of Aden models in Autin et al. 2013).

  - o **Reply:** We added some details on how seeds have been used in previous models. The use of seeds here is somewhat different than in Autin et al. 2013 though (we use linear seeds, where Autin et al. 2013 use broader "patches" of weak materials that cause deformation over a wider area).

- (L143) Methods: I would like to see a bit more discussed on the 4 mm/h velocity since we are in a rotational environment and thus any velocity displays an "angular velocity" character. In line 90 it is mentioned this 4mm/h is the max velocity, so it is worth mentioning that going "south" in the rifted profile means velocities are less.

  - **Reply:** We have added some more context to the text, but we made the choice to define model velocities in mm/h, since these can be directly upscaled and compared to the mm/yr as found in GPS-based studies of plate motions in East Africa (e.g. Saria et al. 2014. Stamps et al. 2021).

- (L143) Methods: Why do you not consider this to be a major issue (answer might be my above comment)?

  - **Reply:** It is not a major issue as long as we do not get wide rifting (distributed deformation due to strong coupling between the foam base and sand layer as a result of too fast movement): the seeds still localize deformation as they are intended to, and the behavior of the model is therefore valid (narrow rifting style).
    - This was already in the following sentences, but recognize that the original manuscript is a bit unclear here, and have rephrased the text to avoid confusion.

- (L172) Methods: agisoft website needs its bracket to close

  - **Reply:** thanks for noticing, it is modified

- (L181-306) Results: There are references to Figures 3-7, such as to the "Malawi Rift", that require the reader to revert back to Figures 1 and 2 to see where that is. Might be worth adding the names of the key features in panel (a) of each of the Figures 3-7.

  - **Reply:** We think this is a good idea and have added the suggested labeling to the figures

- (L331-372 or 415-475) Discussion: See "General comments 1 & 2"

  - **Reply:** We agree and have added details to the introduction and discussion (see reply to previous comments)

- (L423) Discussion: This is the first time that any feature's relative timing is mentioned in the manuscript (see my earlier comment in the Introduction).

  - **Reply:** We agree and have added details to the introduction and discussion (see reply to previous comments)

- Figure 9: Please add scales to the image.

  - **Reply:** We have now added the necessary scales to the figures.

- (L440) Discussion: I would refrain from using the phrase "all things considered" as it is quite informal.

    - **Reply:** We agree and now use "overall" instead.

- (L449-465) Discussion: I recommend using one of the terms "microplate", "micro plate" or "micro-plate" for consistency.

    - **Reply:** Thanks for noticing that the terminology was not consistent, we have opted for using "microplate" throughout the text

**References:**

- Autin, J., Bellahsen, N., Leroy, S., Husson, L., Beslier, M. O., & d'Acremont, E. (2013). The role of structural inheritance in oblique rifting: Insights from analogue models and application to the Gulf of Aden. Tectonophysics, 607, 51-64.

- Farangitakis, G. P., Sokoutis, D., McCaffrey, K. J., Willingshofer, E., Kalnins, L. M., Phethean, J. J., ... & van Steen, V. (2019). Analogue modeling of plate rotation effects in transform margins and rift-transform intersections. Tectonics, 38(3), 823-841.

- Farangitakis, G. P., McCaffrey, K. J., Willingshofer, E., Allen, M. B., Kalnins, L. M., van Hunen, J., ... & Sokoutis, D. (2021). The structural evolution of pull-apart basins in response to changes in plate motion. Basin Research, 33(2), 1603-1625.

- Glerum, A., Brune, S., Stamps, D. S., & Strecker, M. R. (2020). Victoria continental microplate dynamics controlled by the lithospheric strength distribution of the East African Rift. Nature Communications, 11(1), 2881.

- Molnar, N. E., Cruden, A. R., & Betts, P. G. (2017). Interactions between propagating rotational rifts and linear rheological heterogeneities: Insights from three-dimensional laboratory experiments. Tectonics, 36(3), 420-443.

- Phethean, J. J., Kalnins, L. M., van Hunen, J., Biffi, P. G., Davies, R. J., & McCaffrey, K. J. (2016). Madagascar's escape from A frica: A high-resolution plate reconstruction for the W estern S omali B asin and implications for supercontinent dispersal. Geochemistry, Geophysics, Geosystems, 17(12), 5036-5055.

- Reeves, C. V., Teasdale, J. P., & Mahanjane, E. S. (2016). Insight into the Eastern Margin of Africa from a new tectonic model of the Indian Ocean. Geological Society, London, Special Publications, 431(1), 299-322.

- Souriot, T., & Brun, J. P. (1992). Faulting and block rotation in the Afar triangle, East Africa: The Danakil" crank-arm" model. Geology, 20(10), 911-914.

---

## Author Comment (AC2)

**General comments Reviewer 1 – Cynthia Ebinger**

- **Comment:** Zwaan and Schreurs utilize fully scaled analogue models to investigate the role of rotational stresses on the continental-scale spatial evolution of rift zones, with comparison to rifting patterns in East Africa. The models explore a range of co-eval rift initiation geometries, and then track the time evolution of strain for comparison with the modern geometry of active rift zones in parts of E Africa. The models ignore rifting in the offshore SE rift branch in oceanic lithosphere east and southeast of the Rovuma block, and the SW branch that continues perhaps as far west as Angola around the San block.

    - **Reply:** As we now describe clearer in the new version of the manuscript, we implement the **active** plate boundaries for the East African Rift north of the Somalia-Nubia rotation axis, as specified in e.g. Saria et al. (2014) (and in Stamps et al. 2021 too). As such we do include the boundaries of the Rovuma plate in two of our models (A and D), and the eastern boundaries of this plate (which undergo very limited deformation in nature) do in fact show some limited activation in our model D. The latter part is a limitation that is addressed in the text.
    - The SW branch is not included in our models since it is not part of the active plate boundaries as defined by Saria et al. (2014) and Stamps et al. (2021).
    - Note that these southernmost parts of the East African Rift are situated close to the Nubia-Somalia rotation pole, which makes their individual motion rather complex. In our study we focus on the system north of this rotation pole, which is affected by the rotation of the Somalian plate. As such, potential plates further to the south-east (the existence and importance of which are debated) are not included in our models.

- **Comment:** My major concern with the paper is the reliance on outdated structural models based on coarse satellite imagery, and lacking information from seismicity and geodesy that shows modern plate motions. Specifically, the papers cited for the role of pre-existing basement shear zones are all based on satellite imagery and lack the 3rd dimension – the dip of structures. Likewise, the age of some lineaments mapped in these publications remains speculative (e.g., Chorowicz, 2005), and papers utilizing displacement of data strata should only be considered.

    - **Reply:** In our models we implemented the main **active** East African Rift System plate boundaries as proposed by recent authors e.g. Saria et al. (2014), but also in Stamps et al. (2021). This was perhaps not sufficiently clear in the manuscript, and we have now expanded the discussion of weaknesses in East Africa.
    - We do not believe that the dip of structures is of much importance regarding the scale of our study. We aim at modelling the activation of weaknesses and the localization of rifting along the active East African Rift System plate boundaries. The exact type of weakness is of limited important: what is important is that a rift basin forms.

o We agree that the age of the different basins along the East African Rift System remains speculative, as we did point out in our manuscript. We have however expanded the discussion and have added more details on timing of rifting in the introduction as well.

o We do not fully follow the final part of the last sentence.

- **Comment:** The comparisons with data in Section 4.15 as well as model constraints need to be re-thought, and omission of processes (e.g., magmatism, GPE, etc) need to be clearly articulated.

  o **Reply:** We assume the reviewer refers to section 4.2 ("General model limitations") here and have added the suggested processes to the discussion. As specified above, we base our modelling study on the present-day tectonic situation, and explore how rotational motion of the Somalian plate, together with localization of deformation along weaknesses in the lithosphere, can lead to the development of the structures seen in present-day East Africa.

**Suggested literature:**

- **Comment: Seismicity:**

  o (Lindenfeld, M., Rümpker, G., Link, K., Koehn, D., & Batte, A. (2012). Fluid-triggered earthquake swarms in the Rwenzori region, East African Rift—Evidence for rift initiation. Tectonophysics, 566, 95-104

  o Lavayssière, A., Drooff, C., Ebinger, C., Gallacher, R., Illsley-Kemp, F., Oliva, S. J., & Keir, D. (2019). Depth extent and kinematics of faulting in the southern Tanganyika rift, Africa. Tectonics, 38(3), 842-862

  o Ebinger, C. J., Oliva, S. J., Pham, T. Q., Peterson, K., Chindandali, P., Illsley-Kemp, F., ... & Mulibo, G. (2019). Kinematics of active deformation in the Malawi rift and Rungwe Volcanic Province, Africa. Geochemistry, Geophysics, Geosystems, 20(8), 3928-3951

  o Weinstein, A., Oliva, S. J., Ebinger, C. J., Roecker, S., Tiberi, C., Aman, M., ... & Fischer, T. P. (2017). Fault-magma interactions during early continental rifting: Seismicity of the M agadi-N atron-M anyara basins, A frica. Geochemistry, Geophysics, Geosystems, 18(10), 3662-3686

  o Zheng, W., Oliva, S. J., Ebinger, C., & Pritchard, M. E. (2020). Aseismic deformation during the 2014 M w 5.2 Karonga earthquake, Malawi, from satellite interferometry and earthquake source mechanisms. Geophysical Research Letters, 47(22), e2020GL090930)

- **Comment: and geodetic strain from active plate motion and time-averaged strains:**

  o (e.g., DeMets, C., & Merkouriev, S. (2021). Detailed reconstructions of India–Somalia Plate motion, 60 Ma to present: implications for Somalia Plate absolute motion and India–Eurasia Plate motion. Geophysical Journal International, 227(3), 1730-1767

- Stamps, D. S., Kreemer, C., Fernandes, R., Rajaonarison, T. A., & Rambolamanana, G. (2021). Redefining east African rift system kinematics. Geology, 49(2), 150-155;
- Knappe, E., Bendick, R., Ebinger, C., Birhanu, Y., Lewi, E., Floyd, M., ... & Perry, M. (2020). Accommodation of East African rifting across the Turkana depression. Journal of Geophysical Research: Solid Earth, 125(2), e2019JB018469
- Birhanu, Y., Bendick, R., Fisseha, S., Lewi, E., Floyd, M., King, R., & Reilinger, R. (2016). GPS constraints on broad scale extension in the Ethiopian Highlands and Main Ethiopian Rift. Geophysical Research Letters, 43(13), 6844-6851
- Viltres, R., Jónsson, S., Ruch, J., Doubre, C., Reilinger, R., Floyd, M., & Ogubazghi, G. (2020). Kinematics and deformation of the southern Red Sea region from GPS observations. Geophysical Journal International, 221(3), 2143-2154.)

- **Reply:** We kindly thank the reviewer for the suggested literature, and have cited some of these works where appropriate.

- **Comment:** The authors cite decades-old papers to quote a  N to S younging, inferring a N-S propagation of rifting throughout the EAR.  Yet, the only two sectors of the EAR confirming that pattern are the Eastern rift in Kenya and N Tanzania, and the southern Malawi rift.   The MER appears to have propagated from S to N, and the very poorly date Western rift may have initiated near the Rungwe volcanic province, and propagated both N and S from there (e.g., Roberts, E. M., Stevens, N. J., O'Connor, P. M., Dirks, P. H. G. M., Gottfried, M. D., Clyde, W. C., ... & Hemming, S. (2012). Initiation of the western branch of the East African Rift coeval with the eastern branch. Nature Geoscience, 5(4), 289-294.). See also Daly MC, Green P, Watts AB, Davies O, Chibesakunda F, and Walker R (2020) Tectonics and Landscape of the Central African Plateau, and their implications for a propagating Southwestern Rift in Africa. Geochemistry, Geophysics, Geosystems. e2019GC008746. For information on the SW rift zone.

  - **Reply:** The comments are not fully clear, but we believe we can answer here:
  - The evolution of the (various parts of the) East African Rift remains a matter of debate, as we point out in the manuscript. In our study we propose a scenario for this evolution, based on the present-day structures / plate boundaries and the rotation of the Somalian plate. Any deviations from the general pattern we see should then be addressed. We now made sure that this discussion is more extensive, using the highly useful feedback provided by the reviewer.
  - Regarding the propagation of the Main Ethiopian Rift (MER) propagation: this seems a topic of debate (e.g. Bonini et al. 2005; Corti 2009), and is now better discussed in the text, as is the development of the other rift segments.
    - Note however, that there is quite some discussion about the ages of the rift segments, and our model findings are generally in quite good agreement with recent work (e.g. Macgregor, 2015).

o The Western Rift actually propagates in both directions (north and south) in our models, which is in fact in agreement with what the reviewer suggests.

**Other major concerns are as follows:**

- **Comment:** The EAR formed above one or more mantle plumes, and topographic relief within the region is > 1000 km. How do the authors include GPE? If not, tell the reader that it is ignored.

  o **Reply:** The impact of mantle plumes or gravitational potential energy (GPE) is not included in our study. Instead we use a simple set-up to explore the effect rotational rifting due to the rotation motion of the Somalian Plate would have on the evolution of the East African Rift System. We do now address this in the discussion as a potential reason why some of the East African Rift System evolution (as expressed at the surface) could deviate from what the fully rotation-induced rift system (as is simulated in our models would generate). That is, rift initiation timing in the shape of deformation at the Earth's surface could potentially be different, but it is not expected to affect the large-scale rift (deformation) patterns we observe nowadays, which do fit the behaviour of our model C quite well.

- **Comment:** Likewise, parts of the study area have experienced 2+ km of magmatic underplate and have strain accommodated by magma intrusion. Since this is not included, tell the reader and consider the consequences.

  o **Reply:** We do indeed not include magmatic underplating in our models. As suggested by Buck (2004 and 2006), magmatism may promote the localization of deformation. This could impact rift timing and was already mentioned, but is now better explained in the discussion.

- **Comment:** Parts of the study area are underlain by cratonic roots that are 70 km thicker than surrounding areas. Since this is not considered, tell the reader and consider the consequences.

  o **Reply:** It is true that the thickness and rheology of the lithosphere can have a strong impact on the timing of rifting and its expression at the surface. This is now addressed in the text. However, the moment rifting is active along a weakness, plates are in motion and the individual thickness of the different (micro-plates) surrounding these weaknesses is not that important anymore.

- **Comment:** On what basis did the authors choose these models? Without careful consideration of the timing of diachronous rifting data, choice of models could be biased to outdated and inaccurate information.

  o **Reply:** This question is not very clear. We are testing different weakness geometries, based on the situation in the East African Rift System. We believe that the general pattern in our model C fits very nicely with the patters seen

in recent GPS analyses (both Saria et al. 2014 and Stamps et al 2021). Overall, the models nicely show alternative scenarios of East African Rift System evolution that could have emerged if other weaknesses/rift patterns were developed.

- We have now modified the text to better explain the reader what the basis for this modelling study was: we took current plate boundaries as provided by Saria et al. (2014) and Stamps et al. (2021), and combined these with the rotation of the Somalian plate to better understand their impact on the evolution of the East African Rift System.

**Minor comments**

- **Comment:** Segments is generally used to describe a single fault-defined basin unit, whose length depends on the strength of the plate.

    - **Reply:** We believe that the use of "rift segment" is quite ok here.

- **Comment:** Line 48 - Strike slip faulting happens in accommodation zones and may reactivate a variety of basement sttructures, but overall, the rift structures form irrespective of basement structures.  Please don't list my name there.

    - **Reply:** We have rewritten the introduction and the Ebinger et al. 1989 reference is not cited in this context anymore.

- **Comment:** Line 53: What do you mean by an inherited weakness?  THe 'weaknesses' are highly variable in the cited papers.   What about the role of variable rheology linked to plate thickness (and hence geothermal gradient), composition, pre-existing thin zones, availability of melt, not just shear zones:  see Muirhead et al., 2020;

    - **Reply:** These are inherited weaknesses in the lithosphere that allow the easy localization of deformation and the nucleation of rift basins. As the reviewer points out, there are various origins of such weaknesses. We have added some more details to the text to make this clear. But in our models we use a simplified type of weakness in the shape of a "seed", and we believe this is permissible as we focus on large-scale rift patterns and aim to reproduce the arrangement of the East African Rift System.

**Summary**

- **Comment:** The authors have published earlier papers considering rift models, which is in part why I urge them to dig more deeply into assumptions, and to move beyond EAR mythology to factual data constraining modern plate boundary deformation, and to move past lineament analyses.

    - **Reply:** We do not fully follow what the reviewer means with "EAR mythology". It is true that we have published various papers on analogue

models of continental rifting, aiming at better understanding a variety of processes. A number of these modelling papers were parameter studies, but in this manuscript we explicitly aim to apply our models to better understand a specific natural example (the East African Rift System). The result is a model-driven study, providing large-scale insights in the potential evolution of the East African Rift System as the result of the rotational motion of the Somalian plate. This is indeed the great strength of our modelling approach: we can test different scenarios and predict patterns and processes.

o We do agree that there are opportunities to strengthen the links to the natural example along the lines suggested by the reviewer and have done so in the revised text.

References:

- Bonini, M., Corti, G., Innocenti, F., Manetti, P., Mazzarini, F., Abebe, T., Pecskay, Z., 2005. Evolution of the Main Ethiopian Rift in the frame of Afar and Kenya rifts propagation. Tectonics 24, TC1007. doi:10.1029/2004TC001680.
- Corti, G., 2009, Continental rift evolution: From rift initiation to incipient break-up in the Main Ethiopian Rift, East Africa: Earth-Science Reviews, v. 96, p. 1–53, doi:10.1016/j.earscirev.2009.06.005.
- Macgregor, D., 2015..: History of the development of the East African Rift System: A series of interpreted maps through time, J. Afr. Earth. Sc., 101, 232-252, https://doi.org/10.1016/j.jafrearsci.2014.09.016

---

## Author Response (AR2)

Dear Authors,

Many thanks for submitting your revised manuscript, which is improved thanks to the useful comments from reviewers. In particular, amendments made to the introduction, which update the literature basis for EARS inherited structures and extension velocity/propagation, and to the discussion, which make the work clearer/rigorous through detailing additional important limitations of the modelling approach, have improved the work.

In light of the revisions you have made following suggestions from the reviewers, I now recommend your manuscript for publication subject to technical corrections, which I outline below:

- **Reply:** We kindly thank the editor for the positive assessment of our manuscript, and have addressed the comments below

- In line 23 (Abstract), please clarify that 'rotational rifting leads to the overall lateral propagation of deformation towards the rotation access", to be consistent with your reply to Reviewer 1: "The Western Rift actually propagates in both directions (north and south) in our models, which is in fact in agreement with what the reviewer suggests.". This sentence is also slightly long and complex, so I suggest to split it into two.

- **Reply:** We have rephrased the abstract a bit, and made sure to mention that the Western Rift is also seen to be propagating northward, i.e. against the overall southward propagation direction of the East African Rift System. Note that we did a check of the overall manuscript and have slightly modified some wording and corrected figure reference errors here and there, as well as some minor details in the figures.

- Line 59, "comprising" is better changed to "comprises".

- **Reply:** We have corrected the text as suggested

- Line 76, "However" is not necessary here.

- **Reply:** We have removed "however"

- Please take a moment review the written language within your introduction section (which has been subject to numerous revisions) to make sure sentences are now as clear for the reader as they can be. Please also replace some uses of "various" for more specific reference to publications or regions as relevant, which will be more informative for the reader.

- **Reply:** We have rewritten some parts of the text to clarify the characteristics of the East African Rift System. We also did a check of the overall manuscript and have slightly modified some wording and corrected figure reference errors here and there.

- Please add a scale bar to figure 1a.

- **Reply:** We have added a scale bar + lat/lon coordinates to figure 1a, as well as to the similar figure 9a

- Line 297, is the use of "deformation" as you intend in this sentence?

- **Reply:** thanks for noting this, it should be "sieving" instead, and is now corrected

Many thanks,

**Jordan**